# Neural structure of a sensory decoder for motor control

Seth W. Egger 1✉ & Stephen G. Lisberger 1

The transformation of sensory input to motor output is often conceived as a decoder operating on neural representations. We seek a mechanistic understanding of sensory decoding by mimicking neural circuitry in the decoder's design. The results of a simple experiment shape our approach. Changing the size of a target for smooth pursuit eye movements changes the relationship between the variance and mean of the evoked behavior in a way that contradicts the regime of "signal-dependent noise" and defies traditional decoding approaches. A theoretical analysis leads us to propose a circuit for pursuit that includes multiple parallel pathways and multiple sources of variation. Behavioral and neural responses with biomimetic statistics emerge from a biologically-motivated circuit model with noise in the pathway that is dedicated to flexibly adjusting the strength of visual-motor transmission. Our results demonstrate the power of re-imagining decoding as processing through the parallel pathways of neural systems.

[1] Department of Neurobiology, Duke University School of Medicine, Durham, NC 27710, USA. ✉email: seth.egger@duke.edu

The complex circuits of the brain transform sensory inputs into appropriate motor outputs. However, the brain is imperfect, and motor outputs vary from trial to trial even for identical sensory inputs. The variation has a distinct form, leading to well known psychophysical "laws". Fitts law describes a proportional increase in movement variation with the speed or size of the movement[1]. The Weber–Fechner law states that sensory discrimination thresholds grow in proportion with stimulus amplitude[2]. Both laws are classically attributed to "signal-dependent noise", where noise in the brain increases with the amplitude of sensory or motor signals.

Signal-dependent noise has a long history in models of sensory-motor behaviors. To capture the psychophysical laws, models of sensory-motor transformations often take a "black box" approach, where the complete set of operations that transform sensory representations into motor outputs are subsumed into a decoding equation[3–8]. Signal-dependent noise is built into behavioral output either through the properties of the sensory representation[9,10] or noise in the motor command[11,12]. Yet, the responses of neurons in the varied and complex "decoder" circuits between sensory representations and motor coordination vary considerably from trial to trial[13]. The likely presence of decoder noise challenges the "black box" design of decoding models, as the non-sensory, non-motor neurons that perform the computations required of decoding are proposed to contribute to behavioral variation[14,15]. Further, the ability of models that delegate noise exclusively to sensory or motor sources to explain behavioral variance becomes more limited as behavioral tasks become more complicated[16–18]. Our goal is to understand the neural computations that generate sensory-motor behavior, and doing so requires opening the "black box" to consider the computations performed by specific neural pathways as they become engaged during increasingly complex behaviors. The key question that arises from this approach is not whether the responses of neurons in the decoder vary (they do), but rather if the variation that arises in each neural pathway contributes uniquely to behavioral variation as a consequence of the pathway's specific function in sensory-motor transformations.

We approach a mechanistic understanding of sensory decoding by studying the initiation of visually-guided smooth pursuit eye movements, a system where we know much about the underlying neurophysiology. The extrastriate middle temporal visual area (MT) provides sensory signals that drive the behavior[19–21] and several strong empirical observations link the responses of MT neurons to behavioral variation. First, the limits on sensory precision by MT neurons are similar to pursuit eye movements[22]. Second, the properties of MT neurons are appropriate to transform the fluctuations of individual neurons into signal-dependent noise[23]. Finally, trial-by-trial "neuron-behavior" correlations between the responses of individual MT neurons and the eye velocity in the initiation of pursuit argue that correlated sensory noise contributes to motor variation, and places limits on how much variation is added downstream[24]. The robust link between responses in MT and motor behavior has led to a canonical model, where downstream circuits simply decode MT responses and the fluctuations in MT neurons are the primary driver of variation in pursuit.

We start by presenting the simple experimental observation that changing target size breaks the traditional psychophysical laws of signal-dependent noise. The observation was unexpected and cannot be explained by what we know about sensory or motor systems, motivating a consideration of signal and noise in the circuit pathways that make up the sensory decoder. Pursuit eye movements have a neural substrate with at least two pathways radiating from the sensory representation before converging onto the final motor circuits[25]. Anatomically, the output from area MT is transmitted both through fairly direct ponto-cerebellar pathways and through a cortico-cortical circuit that involves the smooth eye movement region of the frontal eye fields (FEF_sem). FEF_sem exerts profound control over pursuit behavior by modulating the strength or "gain" of visual-motor transmission[26]. The "gain control" pathway seems to afford flexible, context-based visual-motor transformations as opposed to reflex-like input–output behavior[25,27–29]. The combination of multiple anatomical, physiological, and behavioral observations leads directly to the proposition that an accurate decoding model will include a pathway for gain control that makes separate and independent contributions to both signal and variation.

We design a sensory-motor decoder to capture our behavioral results. We extrapolate from the extensive data on MT neuron response properties to generate a synthetic population response to each target size and speed. Based on the neuroanatomy of pathways downstream from MT, the tight link between neural activity in MT and FEF_sem[30], and extensive neurophysiological and behavioral data[24,31], our decoder includes parallel pathways for (1) computing the gain of visual-motor transmission and (2) estimating target speed. Under our assumptions about the MT population response and decoding architecture, noise in the gain pathway is required to account for the breakage of traditional psychophysical laws by changes in target size and capture the competing constraints provided by physiological observations.

## Results

### The relationship between behavioral variance and mean changes with the size of the pursuit target.

We start by presenting a simple experiment with results that contradict previous conceptions of the relationship between the variance and mean of motor behavior. We analyzed the eye movements of monkeys that were rewarded for smoothly pursuing patches of moving dots of different sizes. In each trial, a patch of dots centered on the fovea moved within a stationary aperture for 150 ms before moving along with the aperture across the screen. Target speeds were selected randomly from trial-to-trial from 4, 8, 12, 16, and 20 deg/s with equal probability (Fig. 1a, top). Target sizes were selected randomly with equal probability from diameters of 2, 6, and 20 deg (Fig. 1a, bottom). Here, we focus our analysis on the initiation of pursuit, the period before feedback from the motion of the eye can contribute to signal and variation in behavior[32]. Thus, we analyzed the open-loop performance of the visual-motor system to known image motions that were exactly equal to target motion. In this interval, target speed and size strongly influenced mean eye speeds over the first 210 ms after the onset of target motion (Fig. 1b), as expected based on previous experimental results[32–34]. It would not have made sense to analyze variance later in the pursuit response because the impact of visual feedback at later times would cause trial-by-trial variation in the image speed driving later responses.

Our results show that the relationship between the variance and mean of pursuit responses depends strongly and unexpectedly on target size: changing target size disrupts signal-dependent noise. Figure 1c–f plot variance as a function of mean eye speed calculated from trial-by-trial, time-averaged data, using the interval from 110 to 190 ms after the onset of target motion (Fig. 1b, vertical lines). To ensure that latency variation did not contribute to our measures of speed variance, we performed all analyses after using an objective procedure to align all responses by removing trial-by-trial variation in pursuit latency (see "Methods" section). For a given target size, variance increased with the mean eye speed for each individual target size. At a given mean eye speed generated in response to different target sizes, however, variance differed, breaking the relationship expected

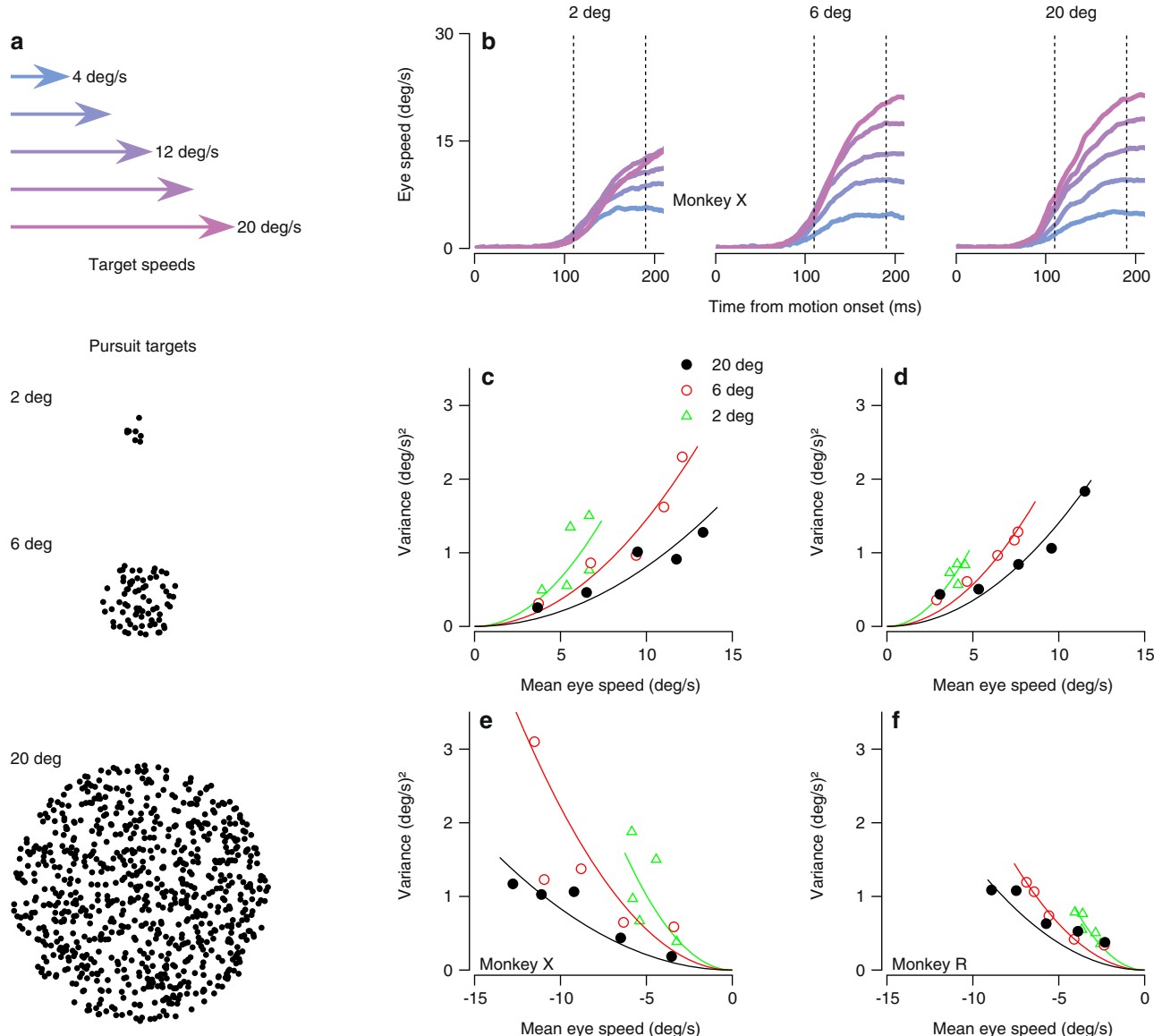

**Fig. 1 Target size affects the relationship between the variance and mean of pursuit initiation. a** Properties of pursuit stimuli. Top: Arrows indicate target speeds. Dot patches moved at a speed selected from a discrete uniform distribution with 5 values evenly spaced between 4 to 20 deg/s (colors). Bottom: Example dot patches corresponding to targets of three different sizes: 2 deg, 6 deg, and 20 deg. **b** Mean pursuit initiation behavior for monkey X, sorted by target speed (colors; see arrows in panel **a**) and patch size indicated at the top of each graph. Each trace shows trial-averaged eye speed as a function of time, starting at the time of target motion onset. Vertical dashed lines indicate the window of time averaging used for subsequent analyses. **c** Variance of eye speed of monkey X plotted against the mean eye speed for rightward pursuit. Symbols plot the behavior for different target speeds and sizes. Curves indicate the fit of a signal-dependent noise model where the Weber fraction is allowed to change with target size. **d** As in panel **c**, but for rightward pursuit in monkey R. **e** As in panel **c**, but for leftward pursuit in monkey X. **f** As in panel **c**, but for leftward pursuit in monkey R. **c–f**, Green, red, and black symbols and curves show results for target sizes of 2, 6, and 20 deg, respectively. Source data are provided as a Source Data file.

based on a standard interpretation of the psychophysical laws. Variance increases more rapidly as a function of mean pursuit initiation speed for the 2 deg patch (Fig. 1c–f; green triangles) compared to the larger patches (red and black circles). The effect of target size on speed variance withstood multiple control analyses, including use of only trials with fixation closer to the center of the patch than required by our fixation window and verification that monkeys pursued the moving target on all trials (Supplementary Fig. 1).

We use a signal-dependent noise model of the form $\sigma^2 = w^2\mu^2$ to quantify the effect of target size on the relationship between mean, $\mu$, and variance, $\sigma^2$. The Weber fraction, $w$, captures the proportionality between mean and variance and allows us to use changes in the

Weber fraction to refer to the effect of target size on signal-dependent noise. We fitted the models to the data for target motion at 4, 12, and 20 deg/s and tested its ability to predict the data for target motion at 8 and 16 deg/s. A model that allowed Weber fractions to vary (curves in Fig. 1c–f; see Supplementary Table 1 for parameter values) better predicted the variance associated with the 8 and 16 deg/s patches across patch sizes by allowing the Weber fraction to decrease with increasing target size (see "Methods" section; fixed $w$ vs. flexible $w$: Monkey X, left pursuit: RMSE = 0.89 vs. 0.76 (deg/s)$^2$, $p = 0.05$, $t(499) = 1.68$; Monkey X, right pursuit: RMSE = 0.63 vs. 0.27 (deg/s)$^2$, $p = 0.01$, $t(499) = 2.20$; Monkey R, left pursuit: RMSE = 0.35 vs. 0.21 (deg/s)$^2$, $p \ll 0.01$, $t(499) = 3.89$; Monkey R, right pursuit: RMSE = 0.46 vs. 0.22 (deg/s)$^2$, $p \ll 0.01$,

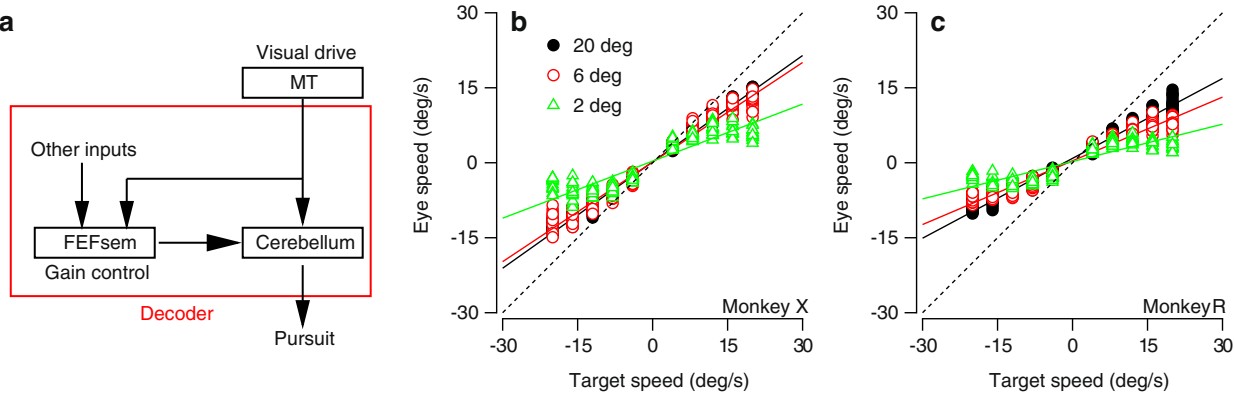

**Fig. 2 Gain of pursuit initiation increases with target size. a** Conceptual circuit model where neurons in the middle temporal area (MT) drive pursuit initiation through (at least) two parallel pathways: a direct pathway to the cerebellum and an indirect pathway for gain control through the smooth eye movement region of the frontal eye fields, FEF$_{sem}$. Arrows indicate the direction of signal flow. The red box indicates that we can think of the entire pursuit circuit downstream of MT as the "sensory decoder". **b** Eye speed during pursuit initiation as a function of target speed for monkey X, sorted by patch size. Each symbol plots the time-averaged behavior for one trial of a given combination of target speed and size. For each condition, we plot a random sample of 15 trials. Lines plot the best fitting linear model to the 2 deg (green), 6 deg (red), and 20 deg (black) targets. **c** As in panel **b**, but for monkey R. Source data are provided as a Source Data file.

$t(499) = 4.61$). The significance of the results were confirmed by a bootstrap analysis using all target speeds where we fitted data from half of the trials and tested the fit with the held-out half of the trials. Also, we observed the same decrease in Weber fractions with target size when we calculated $w$ in 20 ms bins (Supplementary Fig. 2).

Intuitively, one might expect that the change in Weber fraction results from integrating the additional motion information associated with larger target sizes. However, this intuition is valid only when the biological sensors for motion improve their signal-to-noise as a function of target size, or if the downstream decoder can pool information from several sensors with independent sources of noise. While direct measurements of motion responses to the targets used in our task do not exist, several related results argue against either scenario in the case of smooth pursuit. Motion sensitive neurons in area MT, the region of the primate cortex that contributes causally to pursuit[19–21], exhibit decreased signal-to-noise as the size of a motion object extends beyond the classical receptive field[35,36], rather than the increased signal-to-noise required to explain our data. Further, MT neurons are subject to strong noise correlations[37,38], which potentially limit the integration of motion information by downstream decoders of MT activity[39,40].

Neither are our results predicted by standard sensory or motor psychophysical models. Variance in the perception of speed is typically modeled as the result of signal-dependent noise in the sensory system under the assumption that sensory noise increases with speed, $s$, according to $\sigma^2 = w^2 s^2$, where $w$ represents the Weber fraction for perception. However, Weber fractions for speed perception are constant across the range of target sizes used here[41,42], suggesting that the larger number of neurons activated by larger stimuli cannot be used to improve sensory signal-to-noise. Motor noise models represent variance as $\sigma^2 = w^2 \mu^2$, with $w$ now representing a motor Weber fraction. As a result, behavioral variance is tied to the outgoing motor command[11], and standard motor noise models do not predict any difference in the Weber fraction for stimuli that lead to identical mean behavioral output.

**Target size affects the gain of visual-motor transmission.** The fact that target size affects the average eye speed during the initiation of pursuit (Fig. 1b) provides a clue to explanations for the effect of target size on pursuit Weber fractions. Figure 2b, c plot the eye speed of each trial, averaged across the interval from

110 to 190 ms following the onset of visual motion, as a function of target speed separately for each target size. The slope of the relationship between eye speed and target speed increased from 0.31 to 0.52 to 0.62 for the 2, 6, and 20 deg targets in monkey R, and from 0.46 to 0.77 to 0.80 in monkey X (95% confidence intervals were all less than ±0.03). For both monkeys, the slopes associated with the 6 deg target were significantly larger than those for the 2 deg target (monkey R: $z = 34.81$, $p \ll 0.01$; monkey X: $z = 24.11$, $p \ll 0.01$) and those associated with the 20 deg target were significantly larger than those for the 6 deg target (monkey R: $z = 22.45$, $p \ll 0.01$; monkey X: $z = 3.88$, $p < 0.01$).

Considerable prior research on pursuit eye movements allows us to think of eye speed ($E$) at the initiation of pursuit in simple terms as the product of two terms, $E = G\hat{s}$, where $\hat{s}$ is an estimate of target speed based on sensory data before pursuit initiation, and $G$ is the result of a process that controls the strength of visual-motor transmission[43]. Thus, the effect of target size on the slope of the relationships in Fig. 2b, c could represent an effect of target size on $G$ or $\hat{s}$. Available physiological evidence argues against the possibility that the change in slope arises from a change in the speed estimate, $\hat{s}$. When driven by stimuli that extend beyond their classical receptive fields, individual MT neurons exhibit little to no change in speed tuning[44], suggesting the changes in surround stimulation resulting from changes in target size do not alter preferred speeds. Therefore, the computation of $\hat{s}$ should not shift as a function of target size when based on traditional approaches to estimating target speed, such as vector averaging or related algorithms that find the preferred speed at the center of the population response[24,45–47]. Future measurements of MT responses to the stimuli used here are required to validate this assumption.

We propose instead that the effect of target size on the slopes of the relationships in Fig. 2b, c reflects an effect on the gain of visual-motor transmission, $G$. Therefore, we should focus on the neural circuits that control $G$ to explain the companion effects of target size on the Weber fraction of pursuit. Our previous research on the initiation of pursuit has pointed to an overall neural circuit that has two parallel pathways (Fig. 2a). A direct pathway estimates target speed ($\hat{s}$) from the population response in MT. An indirect pathway combines inputs from MT and other inputs to determine how strongly the motor system will respond to a given estimate of target speed by setting the value of $G$. Considerable evidence supports the existence of this second,

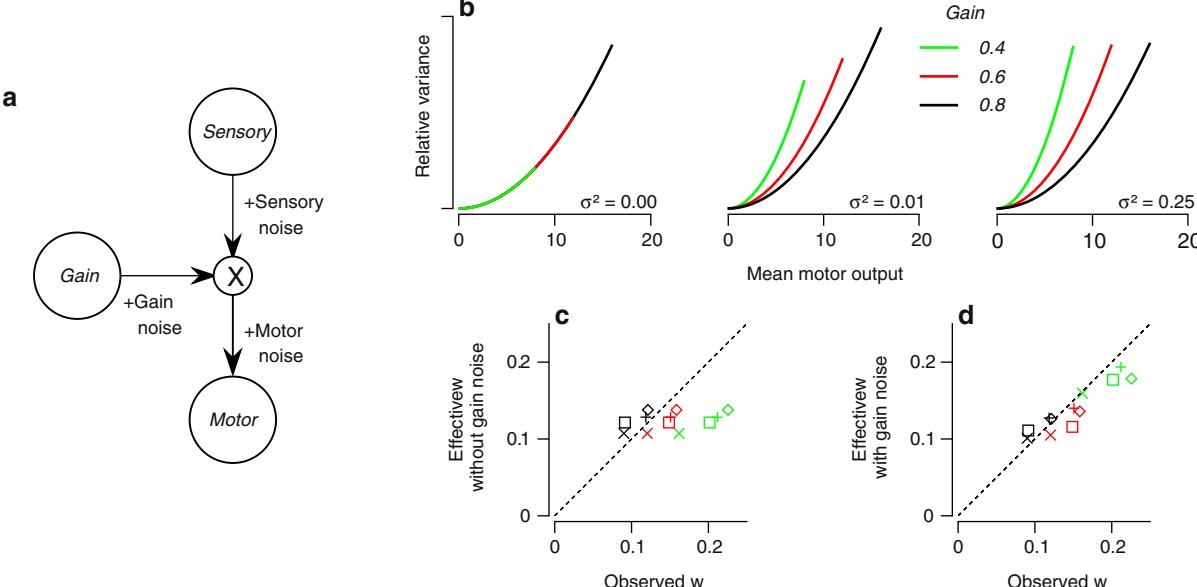

**Fig. 3 The "gain-noise" model: effect of gain noise on the relationship between variance and mean eye speed in behavior. a** Simplified model of sensory-motor transformations, where sensory input is transformed by the application of a gain signal to determine total motor output. **b** Motor variance as a function of mean behavioral output for the simple model (Eq. 1). From left to right, the three graphs show the predictions for gain noise with three different values of variance, $\sigma^2$. Variance is plotted relative to the maximum across gains for each value of noise variance. Green, red, and black curves show predictions for three different mean gains at each value of noise variance. **c** The effective value of the Weber fraction, $w_{\text{eff}}$, inferred from a fit of the simple model without gain noise is plotted against the value of the Weber fraction, $w$, fitted directly to the data for each target size separately (colors correspond to target size, as in Fig. 1c). Plus signs and diamonds indicate right and left pursuit, respectively, for Monkey R. x'es and squares indicate right and left pursuit for Monkey X. Green, red, and black correspond to results from 2, 6, and 20 deg targets, respectively. **d** As in panel **c**, but with gain noise included in the model. Source data are provided as a Source Data file.

"gain-control" pathway[27,29] and assigns this function to the smooth eye movement region of the frontal eye fields or FEF$_{\text{sem}}$[26,30]. Accordingly, we assume changing the size of the target increases the total activity in the MT population response, which increases the gain of visual-motor transmission ($G$) and cause the changes in slope in Fig. 2b, c.

**Noise in the gain of visual-motor transmission captures the observed changes in Weber fractions**. If changing target size alters pursuit initiation by controlling the strength of visual-motor transmission ($G$), then we should look at the pathways that control $G$ as the potential basis for the companion changes in the Weber fraction. Indeed, the challenges of explaining the changes in the Weber fraction in terms of sensory or motor noise sources support a source of noise in the sensory-motor decoder[39].

To develop an intuition for how the pathway that controls visual-motor gain might also influence Weber fractions, we simplified the pursuit system into a computational model that is analytically tractable (Fig. 3a). The model applies a gain, $G$, to the sensory estimate, $\hat{s}$, to generate motor output according to $(G + \eta_G)(\hat{s} + \eta_s) + \eta_m$. The model includes three possible sources of behavioral variation:

1. Sensory noise, $\eta_s$, which we model as Gaussian with variance that increases with sensory signal according to $w_s^2 \hat{s}^2$ to accommodate the increase in perceptual variance associated with increased stimulus magnitude (e.g., Weber's law[2,41]).
2. Motor noise, $\eta_m$, which increases with mean motor output, $\mu$, according to $w_m^2 \mu^2$[11,48].
3. Gain noise, $\eta_G$, which is Gaussian with variance $\sigma^2$.

If the sources of noise are statistically independent, then the mean output, $\mu$, is $G\hat{s}$ and its variance is $(w_s^2 + w_m^2)\mu^2 +$ $\mu^2(\sigma^2 + \sigma^2 w_s^2)/G^2$ (see Supplementary Note 1). Therefore, this simple model produces the standard relationship between the mean and variance, $w^2\mu^2$, where the standard Weber fraction is replaced by the effective Weber fraction in the simple model:

$$w_{\text{eff}} = (w_s^2 + w_m^2) + \frac{\sigma^2 + \sigma^2 w_s^2}{G^2}. \tag{1}$$

We can see from Eq. (1) that the effective Weber fraction, $w_{\text{eff}}$, will depend on $G$ if (and only if) the variance of the gain noise, $\sigma^2$, is non-zero.

Graphical analysis of Eq. (1) confirms that the mean gain becomes an important factor in setting the value of $w$ and therefore the relationship between variance and mean speed in the presence of gain noise ($\sigma^2 > 0$). With gain noise, the variance of motor behavior increases with the mean motor output ($\mu$) at a rate that depends on both the mean sensory-motor gain ($G$) and the variance of gain ($\sigma^2$; Fig. 3b, middle and right graphs). The model predicts that the rate of increase in the variance of motor output will shift in a way that mimics our behavioral data. In the absence of gain noise ($\sigma^2 = 0$), variance grows with mean motor output ($[w_s^2 + w_m^2]\mu^2$) but the model predicts that Weber fractions will be constant across different levels of gain (Fig. 3b, left). While the exact relationship between mean output and variance depends on the linearity of the model (see Supplementary Note 2), the key prediction of a decreasing Weber fraction with increasing mean gain is robust to relaxing the linearity assumption (Supplementary Fig. 3).

The simple model in Fig. 3a fits the change in Weber fraction we observed in pursuit if and only if we allow gain noise. The effective Weber fraction for the model fitted with gain noise agreed with our behavioral observations across monkeys, directions, and target sizes (Fig. 3d, see Supplementary Table 1 for parameter values), whereas the effective Weber fraction for a

model without gain noise did not (Fig. 3c). We further compared the quality of fit of the model with versus without gain noise by fitting models to data from target motions at 4, 12, and 20 deg/s and testing on data from target motion at 8 and 16 deg/s. As before, a model with gain noise predicted the data better than a model that assumed no gain noise when fit to data (predicted variance without vs. with gain noise; Monkey X, left pursuit: RMSE = 0.89 vs. 0.56 $(deg/s)^2$, $p < 0.01$, $t(499) = 2.85$; Monkey X, right pursuit: RMSE = 0.63 vs. 0.44 $(deg/s)^2$, $p < 0.01$, $t(499) = 2.56$; Monkey R, left pursuit: RMSE = 0.35 vs. 0.33 $(deg/s)^2$, $p \ll 0.01$, $t(499) = 3.36$; Monkey R, right pursuit: RMSE = 0.46 vs. 0.32 $(deg/s)^2$, $p \ll 0.01$, $t(499) = 4.81$). Indeed, the relationship between the gain inferred from the behavioral mean and variance alone matched the gain inferred from the trial-by-trial data as a function of stimulus speed (Supplementary Fig. 4). And, again, the statistical significance was confirmed by a bootstrap analysis using all target speeds where we fitted data from half of the trials and tested the fit with the held-out half of the trials.

**Biomimetic model for gain control in pursuit.** The success of a model in predicting a single experimental result is not a strong test of the model, especially in an experimental system like smooth pursuit eye movements where so much is known about the statistics of neural and behavioral responses. Therefore, we have elaborated the model by creating a biologically-realistic MT population response and processing it through a sensory decoder that includes the two known pathways that perform sensory estimation and control of visual-motor gain.

We set the following goals for the elaborate, more "biomimetic" model:

1. The model's signal-dependent noise should shift with target size as in behavior.
2. The variance of the model's output should match the overall variance of behavior.
3. The output of the model should exhibit the correct magnitude of trial-by-trial correlations with the activity of model MT neurons[24].

Our goal was to build a motion representation from model neurons that mimic the statistics of the spike counts of neurons recorded from area MT in response to the onset of target motion and before pursuit is initiated. Each model neuron was selective to a given direction and speed of motion, with mean tuning functions based on the well characterized properties of individual MT neurons and distributed in accord with published data[23,38,49,50] (Fig. 4b). We distributed receptive field centers from 0.25 to 30 deg eccentricity based on the density of MT neurons per degree of visual angle[51,52], we set each neuron's receptive field size based on the known relationship of size with eccentricity[53], and we implemented inhibition outside the classical receptive field according to data from the literature[35,36,44,54,55] (see "Methods" section; Fig. 4d). We generated many single-trial population responses (Fig. 4e) for each target speed and size by (1) measuring mean responses from the model population's tuning curves, (2) adding Poisson-like noise, and (3) constraining the noise with neuron-neuron noise correlations as measured in MT[37–39] (Fig. 4c).

We constructed a sensory decoder that transmits the output of each model MT neuron along two pathways (Fig. 4a), an architecture that accounts for a considerable amount of published behavioral and physiological data[30,56]. The upper pathway performs vector summation of model MT neural responses, following previous results relating motion reliability to the amplitude of the MT population response. It uses that amplitude to set the gain of sensory-motor transmission, $G$[30]. The lower pathway performs vector averaging of the MT population

response to estimate log base 2 of target speed, $\hat{s}$[24,45,47]. We then multiply the outputs of the two pathways ($G\hat{s}$) to complete decoding and generate simulated pursuit behavior. Both of the pathways are needed, for example, to account for the joint effects of target speed and contrast on eye speed in the initiation of pursuit[30,57]. Indeed, the non-linearity created by decoding the log of speed is compensated by the non-linearity of the gain increase as a function of speed, and allows the mean output of the model as a function of target speed to match the data qualitatively.

Because larger motion patches increased the number of model MT neurons activated by the stimulus (Fig. 4d), vector summation performed by the motion reliability pathway generated a gain that increased with patch size when averaged across trials (Fig. 4f). Gain also increased with target speed due to the log-normal shape of the speed tuning of MT neurons and the fact the weighting in the model increases with preferred speed (Fig. 4f). Multiplication of the output of the gain pathway times the target speed provided by the vector averaging speed estimation pathway resulted in mean eye speeds similar to those observed in our monkeys (Fig. 4g). Specifically, the decoder's output generated mean eye speeds that increased steeply as a function of target speed at a rate that depended on target size.

**Noise in the gain-control pathway reproduces the statistics of behavioral and neural data.** When we add noise in the gain-control pathway, our biomimetic model predicts all three of the key statistics we set out to reproduce: an effect of target size on Weber fractions, realistic magnitudes of variance in eye speed at the initiation of pursuit, and realistic trial-by-trial correlations between the activity of individual MT neurons and pursuit eye speed. Without noise in the gain-control pathway, the model failed to predict any of these statistics.

Comparison of Fig. 5b, c shows that noise in the gain-control pathway allows the rate of increase in model variance to depend on the size of the target: the variance in the simulated response to the 2 deg patch (green triangles) increases more rapidly than for the 6 deg patch (red circles), with a still slower rise for the 20 deg patch (black circles). The presence of gain noise also increases the magnitude of the variance in eye speed so that it matches more closely the values in the data from our monkeys.

Comparison of Fig. 5d–f shows that noise in the gain pathway allows the model to reproduce the MT-pursuit correlations recorded by Hohl et al.[24]. To obtain the data in Fig. 5d, the authors recorded up to 500 responses of individual MT neurons to the same moving target during the initiation of pursuit. They then computed trial-by-trial correlations between the spike count of the MT neuron and the eye speed in the initiation of pursuit, and plotted the result as a function of the relationship between the target speed and the preferred speed of the neuron. They found statistically significant, mostly positive correlations in many neurons (Fig. 5d).

We compared the results of our model to the data by analyzing model MT-pursuit correlations while limiting the neurons studied to those with preferred or antipreferred directions near that of the target direction, as had been done in the experimental study (see "Methods" section). Without gain noise (Fig. 5e), the model predicts MT-pursuit correlations that are mainly positive, as in the data, but the magnitude of the correlations is considerably larger than in the data (Fig. 5d). Gain noise is effectively "noise added downstream" and therefore reduces the magnitude of the model's MT-pursuit correlations (Fig. 5f) to a level closer to the measured values. We also note that the 2-pathway decoder used in our model (Fig. 5a) reproduces the lack of dependence of MT-pursuit correlations on target speed relative to preferred speed. This is in contrast to simple vector

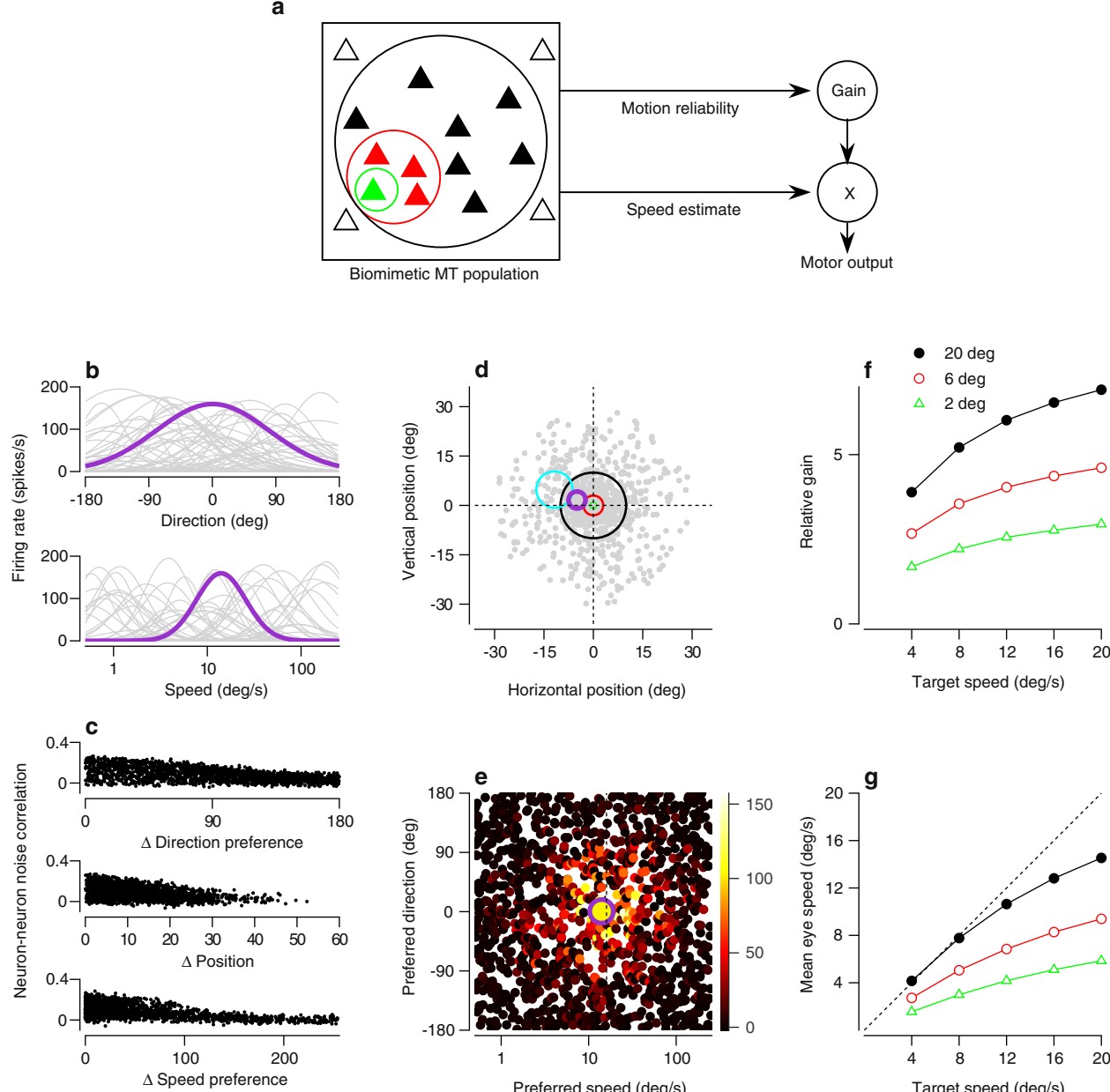

**Fig. 4 A more biomimetic model of MT and the downstream pursuit circuit. a** Overview of the biomimetic circuit model of pursuit. Activation of model middle temporal (MT) neurons (triangles within square, left) depends on patch size (colored circles). Each neuron drives (1) the motion reliability pathway, which computes a vector sum of MT activity (top) and (2) the speed estimation pathway, which computes a vector average (bottom). Application of the output of the motion reliability pathway to the output of the estimation pathway generates simulated pursuit (right). **b** Direction tuning (top) and speed tuning (bottom) of a random sample of model MT neurons. Purple highlights one neuron. **c** Noise correlations between model neurons as a function of the difference in direction preference (Δ Direction preference; top), distance between receptive field positions (Δ Position; middle), and difference in speed preference (Δ Speed preference; bottom). Each dot plots the measured noise correlation for a different pair of model neurons sampled from the population. **d** Receptive field centers (gray points) of the population of MT neurons used to represent motion for pursuit. Green, red, and black circles correspond to the locations of the 2 deg, 6 deg, and 20 deg targets. Purple and cyan circles plot the classical receptive field of two example neurons. **e** Response of each model neuron for an example trial (spikes/s; color). The center of each point corresponds to the neuron's preferred direction and speed. Large purple symbol indicates the example neuron from panels **b** and **d**. **f** Mean gain of the population as a function of target speed and size. **g** Mean simulated eye speed as a function of target speed and size (colors as in panel **f**). Source data are provided as a Source Data file.

averaging, which predicts negative correlations for target speeds higher than the preferred speed of a given neuron[24]. Overall, the presence of gain noise has the expected effect of adding downstream noise. It both increases the variance of model output and partially decorrelates the responses of MT neurons from model output[15].

While our circuit leverages much of what is known about MT response properties, many detailed aspects of the model remain poorly constrained by results from neurophysiology. Therefore, we parameterized surround suppression and threshold nonlinearities in our model units and tested the impact of each on our results. Across a wide range of parameterizations, no instantiation

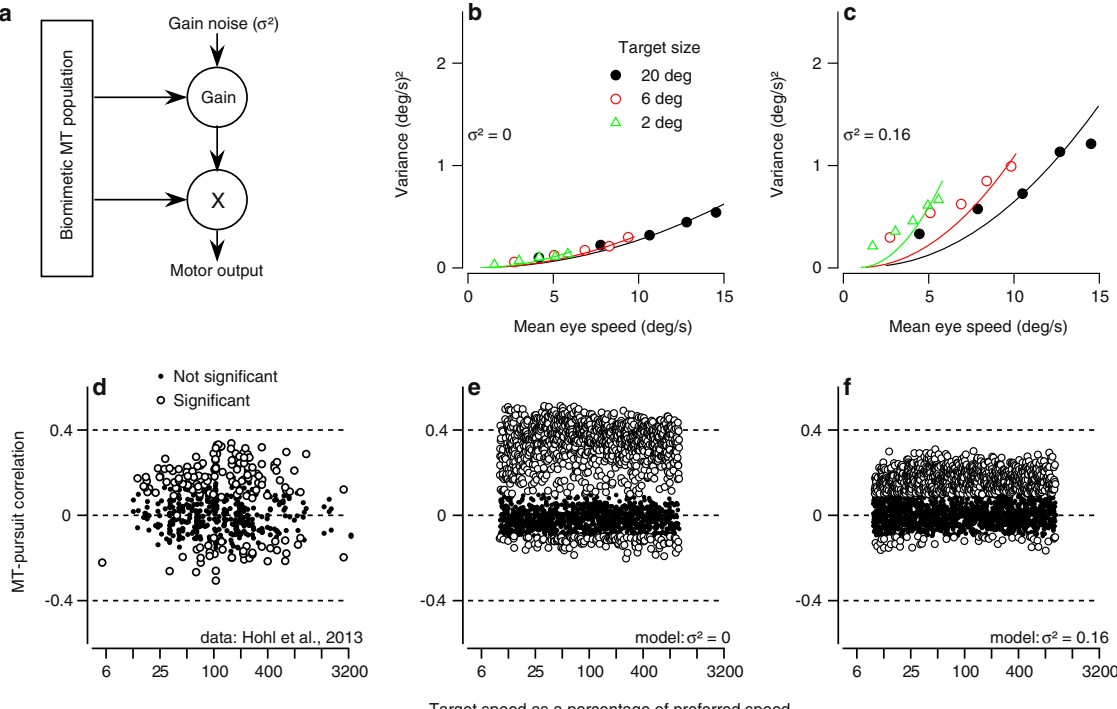

**Fig. 5 Biomimetic model with gain noise captures key features of behavioral and neurophysiological data. a** Biomimetic model as in Fig. 4a, but highlighting noise added to the gain pathway downstream of middle temporal (MT) neurons. Noise was drawn from a zero-mean Gaussian with variance $\sigma^2$. **b** Relationship between variance and mean eye speed of model output without gain noise. Green triangles, red circles, and black circles correspond to model behavior in response to 2, 6, and 20 deg targets, respectively. Green, red, and black curves plot fits of the gain-noise model from Eq. (1) to circuit model responses to 2, 6, and 20 deg targets, respectively. **c** As in panel **b**, but with gain noise. **d** Observed MT-pursuit correlations, reproduced from Holh et al.[24]. Each symbol plots the trial-by-trial neuron-behavior correlation for an individual neuron. Open circles indicate statistically significant correlations ($p < 0.05$). **e** MT-pursuit correlations measured for the model neurons in the biomimetic model without gain noise when the patch size was 20 deg. Conventions as in **d**. **f** As in panel **e**, but with gain noise. Note that for both the data and the model, the points in **d–f** with values of MT-pursuit correlation less than zero came almost entirely from neurons whose null direction was aligned with the direction of target motion. Source data are provided as a Source Data file.

of the circuit recapitulated the effect of target size on Weber fractions and predicted realistic amplitudes of behavioral variance without noise in the gain-control pathway (Supplementary Fig. 5). Similarly, the change in Weber fraction and magnitude of MT-pursuit correlations could not be captured by simply increasing the size of the simulated population without adding gain noise (Supplementary Fig. 6). Still, the response properties measured from stimuli optimized to the receptive fields of individual MT neurons may not generalize to the conditions of our behavioral experiment. Our results might be explained by signal and noise in MT alone if as yet unmeasured MT response properties (i) support motion integration during pursuit beyond that which occurs perceptually[41,42] and (ii) revealed higher magnitude noise correlations in MT than what is expected based on previous measurements. We think it is unlikely that both of these conditions will be met, but revisions to our model would be necessary if future experiments reveal errors in our assumptions.

## Discussion

Behavior results from the complete set of operations that transform sensory representations into motor output, which we refer to here as sensory decoding. There are many examples where application of decoding equations to sensory representations have elucidated how the brain generates motor or perceptual behavior[3–8]. However, to accommodate the complexity of the natural behavioral repertoire, a decoder requires the capacity to flexibly map sensory inputs to motor outputs. By incorporating the known structure of the neural system for smooth pursuit into

a decoder and leveraging structural noise, we have created a flexible sensor-motor circuit that is biologically more realistic and that reproduces the complete statistics of sensory-motor behavior more powerfully than traditional decoding equations. Our analysis demonstrates how the conventional decoding approach to understanding the neural basis for perceptual or motor behavior can be re-imagined and aligned better with biological reality.

We have combined theory, measurements of the statistics of behavior and neural responses, and explicit modeling of the functional pathways of the sensory-motor decoder. Our approach was driven by extensive knowledge of the relationship between the functional components of visually-guided smooth pursuit eye movements, and firing properties of neurons in the relevant circuits. Recordings made in monkeys during pursuit eye movements have led to a strong hypothesis for the structure of this sensory-motor system[30,43] and focused us on a sensory decoder with two pathways and two functional components. Both pathways rely on inputs from the sensory representation of visual motion in area MT. The first pathway estimates target speed, regardless of target size, by performing vector averaging of the MT population response. Vector average decoders successfully capture much of pursuit behavior[4,24,38,47], and have the critical emergent property that they predict larger magnitude pursuit initiation for faster target speeds. The second pathway computes the gain of the sensory-motor transmission based on the fact that the evoked eye movement is modulated by the context of a visual motion input[27–29]. The responses of neurons in FEF$_{sem}$ substantiate a 2-pathway architecture that includes gain control[58–60]

and FEF$_{sem}$ has a causal relationship with the gain of visual-motor transmission[26,61,62].

Building a two pathway structure into the model sensory decoder allowed us to capture our observation that changing target size also changed the Weber fraction for the initiation of pursuit. The two-pathway structure allows for a plausible source of independent noise in gain control that captures the changes in Weber fraction we observe. The resulting model's output reproduced the measured amplitude of behavioral variation and also matched the trial-by-trial correlations between model MT responses and pursuit behavior to those observed in physiology. Thus, a decoder with multiple functional pathways and a finite contribution of noise provides a parsimonious account of a wide-ranging set of results.

Previous publications have concluded that the variation in sensory-motor behavior could be explained in terms of noise in the sensory system alone for both pursuit and other motor behaviors[63,64]. However, increasing evidence from both behavior[16–18] and physiology[14,65,66] suggests that neural systems between sensation and action, including gain systems[31,67], are subject to noise that propagates into motor output. Under standard experimental conditions, the expected impact of gain noise on behavioral variation is identical to that of sensory noise and so the analytical results of Osborne et al.[63] would emerge whether behavioral variation arises from sensory noise only, or from a combination of sensory and gain noise. Thus, our conclusion that the computation of sensory-motor gain is subject to an independent source of noise that arises from gain control does not contradict previous data, only previous conclusions.

Several lines of evidence argue against the logical idea that integration of the additional information associated with increasing target size should decrease the effective Weber fraction in behavior[68], and therefore explain our behavioral results. First, due to surround suppression[35], the average MT neuron exhibits decreased precision as target size increases beyond the classical receptive field[36], not the increased precision that would be required to explain our data. Second, increasing target size does not lead to increased precision during perceptual decision making that is based on the same sensory representation that drives pursuit initiation[41,42]. Finally, the correlated variability of MT neurons appears to limit the increase in information with increasing population size[38,39]. Indeed, our model's inclusion of correlated variability in MT neurons with different receptive field locations provides a parsimonious account of results from experiments with both motor and perceptual endpoints.

This final point, however, deserves additional consideration given theoretical results demonstrating that noise correlations do not necessarily limit information recovered by the downstream decoder. Whether noise correlations harm or help in decoding depends on the tuning functions of the population, the structure of the noise correlations, and the form of the decoder[69,70]. It has been shown that, for a model of MT with realistic tuning properties, an optimal decoder can "average out" correlated noise to recover speed with precision limited only by neuron number, assuming the noise correlations decay with the difference in preferred speed and no other structure exists[40,71]. However, the noise correlations in MT also depend on other stimulus features (e.g., direction preferences and the distance between receptive fields[38]), and each additional source of correlated variability decreases the likelihood that an optimal decoder can average out noise[40]. Further, it is an open question if the decoder of MT activity is optimal. A given linear decoder will only optimally estimate speed from MT neurons over a limited range of target speeds, contrasts, and forms[40]. Achieving an optimal estimate across the range of natural motion inputs requires stimulus-dependent adaptation of the linear decoder. Implementing a flexible gain to modulate stimulus estimates, as modeled in our biomimetic circuit, may reflect a decoding solution sufficient for pursuit initiation, obviating the need for an optimal decoder. Future experimental and theoretical work is required to characterize the structure of noise correlations and the construction of the pursuit decoder to provide a full account of variability in pursuit initiation.

MT neurons drive the initiation of pursuit[19–21]. Therefore, it was important in creating our model MT population response to assess critically the signal and noise properties of MT neurons in response to our stimuli. One important consideration is how stimuli outside the classical receptive fields of MT neurons influence the representation of motion by the population. We modeled our neurons with a uniform surround that divisively inhibits the response of the classical receptive field[35,36,47]. Under this assumption, made because surround stimulation changes the amplitude but not other properties of MT neuron speed tuning[44], our decoding model is robust to the degree of suppression by surround stimulation. However, extra-classical surrounds of MT neurons are varied and complex[72], and a still more accurate model of how our stimuli impact the responses of MT neurons and downstream systems would be the next step to completely explain the complex spatial integration properties of pursuit[73]. Another critical assumption is the noise properties of MT itself. We have modeled Poisson-like noise that is consistent with classical measurements from MT[74,75]. However, recent results have demonstrated that the variability of MT neurons can be partitioned into Poisson-like noise and gain noise that is correlated across neurons[76]. Therefore, while we have modeled noise in a feed-forward pathway through FEF$_{sem}$, it is possible that gain signals with noise are fed back to MT during our task.

It is also important to consider how the many other pursuit related areas[77] outside of MT, FEF$_{sem}$, and the cerebellum might contribute to signal and noise of pursuit in our task. Larger stimuli increasingly engage neurons in the medial superior temporal cortex (MST)[78], an area implicated in the reflexive pursuit in response to large-field motion stimuli known as ocular following[79]. While stimuli used to drive both MST neurons and ocular following are typically larger than those used here, we cannot rule out the possibility that the changes in gain we observed can be attributed to an increased ocular following based on the MST pathway. Finally, while motion responses in MST and other areas are typically thought to be driven by MT, it is possible that our stimuli activate alternative motion pathways with signal and noise properties that can explain our results. We stress that, although made quite specific here, our results apply generally to decoders of any architecture that implements the computation $f(\mathbf{r}_{motion})\mathbf{w}^T\mathbf{r}_{motion}$, where $\mathbf{w}$ is a vector of decoding weights, $\mathbf{r}_{motion}$ is a vector of motion responses, and $f(\mathbf{r}_{motion})$ returns a gain with noise that is independent of the noise in $\mathbf{r}_{motion}$. Through this framework it will be possible to test the contributions of MT, FEF$_{sem}$, MST, or other areas to the effects we observe.

Our choice of decoding architecture is strongly informed by previous work that highlights the importance of stimulus reliability to motion processing. Previous results have demonstrated that observers tend to have a bias in speed perception toward slower speeds that depends on stimulus contrast[80–82]. A similar result applies to smooth pursuit, where the motor output shows systematic biases that also depend stimulus contrast[28,29]. Assuming stimulus reliability increases with contrast, these results can be explained by a strategy that performs a reliability-weighted combination of the estimate of the sensory stimulus with prior expectations[28,29,82]. A similar argument can be applied with regards to target size, where reliability, and therefore stimulus weight, increases with target size. Indeed, previous work from our

lab has identified FEF$_{sem}$ as a critical locus for computing a reliability weighted sensory-motor gain based on motion contrast[30], and the modeling here applies that architecture to pursuit in response to targets of different size.

Decoding architectures that employ gain control apply broadly to sensory-motor behaviors. The reliability-based adjustments to estimation discussed in relation to speed have been observed in behavior across modalities and tasks[68,83–89]. These results support a general computational strategy of sensory estimation by Bayesian integration[90]. Implementation of a reliability-weighted computation can be achieved by the appropriate application of sensory gains, suggesting a general role of gain control during estimation behavior. Gain control also plays a role in optimizing motor behavior[91,92]. Here, the flexible application of gain on sensory estimates modulates control efforts that are relevant to completing the task[12,93]. Given the importance of gain control to sensory-motor behaviors, it is perhaps not surprising that dedicated pathways exist in neural circuits to learn and implement context-specific sensory-motor gains[30]. Our findings suggest that the flexibility afforded by independent gain control pathways comes at a cost of increased variability not typically accounted for by optimization models. Future models that include gain noise may better capture sensory-motor policies implemented by the brain[67].

More generally, our results have broad applicability to models that attempt to explain signal and noise in complex behaviors. The brain's computational power is achieved through numerous pathways that process information in parallel before ultimately driving motor output. Our results stress the importance of understanding how noise in each of the processing pathways contributes to behavioral variation. If we can characterize and model the pathways in sensory decoders for movement and identify noise specific to each neural pathway, then we can constrain the possible circuit implementations and generate specific, experimentally-testable predictions for circuit function.

## Methods

Two male rhesus macaques, aged 12 and 14 years and weighting between 12.0 and 16.8 kg, performed the pursuit task. All experimental protocols were approved by the Institutional Animal Care and Use Committee at Duke University (protocol number A085-18-04) and performed in accordance with the National Institutes of Health Guide for the Care and Use of Laboratory Animals. Eye movements were tracked with a scleral search coil while their heads were fixed by a restraint post[94]. Implantation of experimental apparati was performed using sterile procedures under general anesthesia with isoflurane. After implantation, animals were trained to fixate and pursue visual targets for juice reward based on the analog signal from the implanted search coil. Both animals had extensive experience performing pursuit tasks before this experiment.

**Stimuli and task.** Visual targets were displayed on a 23″ CRT monitor with an 80 Hz refresh rate at a distance of 30 cm. At this distance the screen subtended 77 deg of visual angle horizontally and 55 deg vertically. Stimulus presentation and timing were controlled by custom software (Maestro).

Each trial began with the monkey fixating a white, 0.5 deg, circular fixation point for a random interval selected from a uniform distribution between 300 to 400 ms. After the monkey maintained fixation within a 3.5 deg radius around the target for the entire interval, the fixation point disappeared and was replaced by a target for pursuit. Results were unchanged if we analyzed data from trials with gaze directions within 2 (monkey R) or 2.5 (monkey X) deg of the fixation target. Targets were dot patches presented in an invisible circular aperture with a diameter selected each trial uniformly from one of three values: 2, 6, or 20 deg. All targets were 100% contrast, white dots on a black screen, with dot density set to 2.55 dots/ deg$^2$ and the placement of the dots randomized across trials. Upon presentation, dot patches moved either to the right or left within their aperture for 150 ms before continuing to move globally across the screen for 750 ms. This procedure reduced the occurrence of catch up saccades during pursuit initiation[22]. Patches moved at a speed selected for each trial from a discrete, uniform distribution with 5 speeds linearly spaced from 4 to 20 deg/s. Monkeys were rewarded for maintaining their eyes within 3.8, 3.8, or 10 deg of the center of the visual target throughout the presentation of the 2, 6, and 20 deg diameter targets, respectively; trials were aborted if the monkey's eye strayed outside of this window. After either successful completion or an aborted trial, the fixation point for the next trial was presented immediately.

**Data analysis.** Horizontal and vertical eye position and speed for each trial was stored for offline analysis. Monkey R and monkey X completed 4027 and 1442 total trials, respectively. For each trial, we first detected the presence of saccades within a 250 ms window starting from motion onset. Any trial with a saccade was discarded from future analysis. Both monkeys reliably initiated pursuit in these trials, with monkey X and monkey R reaching 20% of target speed by 190 ms after motion onset in 100% and 99.5% of trials, respectively. Visual inspection of the remaining 0.5% of trials from monkey R revealed pursuit was initiated with a long latency or small gain. To eliminate variation due to trial-by-trial differences in pursuit latency, we used a previously published algorithm to infer the time of pursuit onset relative to the mean behavior[31]. Briefly, we averaged horizontal eye speeds, conditioned on the target speed and size, across trials in a window 0 to 250 ms after motion onset. We then shifted and scaled traces from each trial until they best matched the conditional mean trace by searching a grid of time-lags between −40 and 50 ms, gains between 0.2 and 1.9, and speed offsets between −2 deg/s and 2 deg/s. After realigning the horizontal velocity trace for each trial to mean pursuit onset, we averaged the speed on each trial within a window 110 to 190 ms relative to target motion onset and then computed statistics across all trials with identical targets and target motions. Subsequent analyses were performed using the time-averaged data.

To test if Weber fractions associated with the initiation of pursuit changed with target size we determined the mean and variance of pursuit speeds, conditioned on target speed, patch size, and motion direction. We then fit the behavioral data to two models to each motion direction, both of the form $\sigma^2_{pre_{i,j}} = w_i^2 \mu_{i,j}^2$, where $\mu_{i,j}$ was the mean of the observed behavior, $i$ indexes target sizes, and $j$ indexes target speeds. Under the null hypothesis we held $w_i$ fixed across target size. Under the hypothesis that Weber fractions changed, we allowed $w_i$ to vary across target sizes. For each model, we minimized the following equation:

$$SSE = \sum_{i=1}^{M} \sum_{j=1}^{N} (\sigma^2_{obs_{i,j}} - \sigma^2_{pre_{i,j}})^2, \tag{2}$$

where $M$ is the number of different patch sizes in a training set, $N$ is the number of different target speeds in a training set, $\sigma^2_{obs_{i,j}}$ is the observed variance for the $i$th patch size and $j$th target speed, and $\sigma^2_{pre_{i,j}}$ is the corresponding variance predicted by either hypothesis. For each model we minimized the sum of squared errors across all patch sizes and target speeds of 4, 12, and 20 deg/s. To test which model better fit the data, we computed the root mean squared error (RMSE) in the predicted variance of each model across patch sizes for target speeds of 8 and 16 deg/s. RMSE was defined as

$$RMSE = \sqrt{\frac{1}{MN} \sum_{i=1}^{M} \sum_{j=1}^{N} (\sigma^2_{obs_{i,j}} - \sigma^2_{pre_{i,j}})^2}. \tag{3}$$

To evaluate the significance of the improvement of fit we sampled 50 percent of the trials for each condition and used these data to fit each model. We then found the differnce in RMSE between the models on the remaining trials. We repeated this 1000 times to generate a bootstrap distribution of RMSE differences and then calculated the $t$-statistic as the mean divided by the standard deviation of the bootstrap distribution. The $t$-statistic was used to perform a paired $t$-test to determine the probability that the distribution of RMSE differences was consistent with the null hypothesis that the models had equal RMSEs. We chose to use 50% of trials to maximize the reliability of variance estimates for both training and test data.

We performed a similar analysis to test if the data were consistent with gain noise. We fit two models to the data: the gain noise hypothesis, $H_{\sigma > 0}$, or the null hypothesis without gain noise, $H_{\sigma = 0}$. For each model, we minimized Eq. (2), except we set

$$\sigma^2_{pre_{i,j}} = w_s^2 \mu_{i,j}^2 + \frac{\sigma^2 + \sigma^2 w_s^2}{G_i^2} G_i^2 s_j^2. \tag{4}$$

Because the term $G_i$ cancels, $w_s$ and $\sigma$ are the only free parameters for the gain noise hypothesis. For the null hypothesis, $w_s$ was a free parameter and $\sigma$ was set to 0. We followed the same procedure as above, fitting the data across patch sizes and target speeds of 4, 12, and 20 deg/s and testing the fit by calculating the RMSE according to Eq. (3) across patch sizes for target speeds of 8 and 16 deg/s. As above, significance was determined through a bootstrap analysis. To estimate effective Weber fractions under the gain-noise model, we used the fit values of $w_s$ and $\sigma$ for each model, and $G_i$ was found via linear regression of the eye speed vs. target speed data (i.e., Fig. 2b, c). Fitting a model which included an additional parameter for motor noise (i.e., $w_m$ in the main text) did not change the results (Supplementary Table 1), which was expected because motor noise will not change the effective Weber fraction for different target sizes and the effect of $w_m$ and $w_s$ on predicted variance is virtually identical in the model when $w_s$ and $\sigma$ are less than 1.

**Biomimetic circuit model of pursuit.** Our circuit model of pursuit was based on a population of model MT neurons with realistic tuning properties. Based on estimates

of the cortical size of the representation as a function of eccentricity in mm[251,52], we modeled the density of neurons as a function of eccentricity as uniformly distributed near the fovea (eccentricities between 0.25 and 1 deg) and as 6*eccentricity$^{-0.9}$ for eccentricities greater than 1. We than randomly sampled receptive field centers from this density, assuming 180 foveal neurons and the remaining neurons spanning eccentricities of 1 deg to 30 deg. Results reported in the main text model 1280 MT neurons within this range. This number of neurons was chosen to balance computational demands with the effect of population size on overall variance, changes in Weber fractions, and MT-pursuit correlation strength. Our results do not critically depend on the exact number of MT neurons simulated (Supplementary Fig. 6).

We modeled the response to a target of direction $\theta$ of each MT neuron, indexed by $i$, as

$$f_\theta^i(\theta) = e^{-\frac{(\theta-\theta_i')^2}{2\tau_{\theta_i}^2}},$$ (5)

where $\theta_i'$ is the preferred direction, and $\tau_{\theta_i}$ determines the width of the response function. $\tau_{\theta_i}$ was sampled at random for each neuron from a uniform distribution between 20 and 90. The preferred direction was sampled from a uniform distribution between $-180$ and 180 deg.

Each model neuron's response to a target of speed $s$ was

$$f_s^i(s) = e^{-\frac{(\log_2 s/s_i')^2}{2\tau_{s_i}^2}},$$ (6)

where $s_i'$ is the preferred speed, and $\tau_{s_i}$ determines the width of the speed response in logarithmic space. $\tau_{s_i}$ was sampled from a uniform distribution between 0.64 and 2.8. The preferred speed was sampled from a uniform distribution over $\log_2(s)$ with $s$ between 0.5 and 256 deg/s.

Finally, we modulated the response based on the overlap the dot stimulus had with the receptive field of each neuron. Based on previous measurements[53], we assumed the receptive field diameter grew with the eccentricity according to $(0.69 * \text{eccentricity} + 1)/\sqrt{\pi}$. We then determined the response based on stimulation of the classical receptive field as

$$r_{crf}^i = \left(\left[\frac{\sqrt{z^i(\rho)} - \omega}{1 - \omega}\right]^+\right)^n,$$ (7)

where $z^i$ takes a value between 0 and 1 that represents the fraction of the classical receptive field overlapped by the stimulus of size $\rho$. The symbol $[x]^+$ represents positive rectification of the value $x$, allowing $\omega$ and $n$ to model threshold nonlinearities in receptive field summation by MT neurons[95,96]. Values of $n$ near 1 will produce responses that are close to receptive field summation. Values approaching 0 allow the response of a neuron to approach its maximum whenever the stimulus partially overlaps with the classical receptive field.

Previous experiments have further documented that stimuli outside the classical receptive field of MT neurons tend to suppress the response to stimuli within the receptive field[35,72,97]. We assumed an extra classical surround with a radius three times the size of the classical receptive field. After calculating the overlap of the stimulus with the extra classical surround, $z_{sur}^i$, the response of the surround was

$$r_{sur}^i = \left(\left[\frac{\sqrt{z_{sur}^i(\rho)} - \omega}{1 - \omega}\right]^+\right)^n,$$ (8)

with the threshold nonlinearity identical to that used for the classical receptive field. We then modeled surround suppression as divisive normalization such that the response due to the receptive field properties was

$$f_\rho^i(\rho) = \frac{r_{crf}^i}{(1 - \beta) + \beta(\epsilon + [r_{crf}^i + r_{sur}^i]/2)},$$ (9)

where $\beta$ determines the degree of surround suppression, and $\epsilon$ is a constant which normalizes the response to be between 0 and 1 when $\beta = 1$. For each of the figures presented in the main manuscript, we set $\omega = 0$, $n = 1$, and $\beta = 1$.

Putting together the response due to direction, speed, and receptive field position, the deterministic response of each neuron was then modeled as

$$f^i(\theta, s, \rho) = A_i f_\theta^i(\theta) f_s^i(s) f_\rho^i(\rho)$$ (10)

where $A_i$ specifies the overall amplitude of responses in spikes/s and was sampled at random for each neuron from a uniform distribution between 20 and 200.

To emulate the noise properties of MT neurons, we added Poisson noise that was correlated across neurons. Following[38], we modeled the correlated noise as

$$r_{i,j} = r_{max}\left[e^{-\frac{\Delta PD_{i,j}^2}{\Delta PD_{max}^2 \lambda_\theta^2}}\right]\left[e^{-\frac{\Delta PS_{i,j}^2}{\Delta PS_{max}^2 \lambda_s^2}}\right]\left[e^{-\frac{\Delta C_{i,j}^2}{\Delta C_{max}^2 \lambda_C^2}}\right],$$ (11)

where $i$ and $j$ index neurons, $r_{i,j}$ is the noise correlation between the pair, $r_{max}$ set the maximum value of the correlation, $\Delta PD_{i,j}$ is the difference in preferred motion direction, $\Delta PS_{i,j}$ is the difference in preferred speed, $\Delta C_{i,j}$ is the distance between receptive field centers, $\Delta PD_{max}$ is the maximum difference in preferred directions, $\Delta PS_{max}$ is the maximum difference in preferred speeds, $\Delta C_{max}$ is the maximum distance between receptive fields. $\lambda_\theta$, $\lambda_s$, and $\lambda_C$ set the rate of decay in correlation

with increasing difference in preferred distance, speed, and receptive field position, respectively. We set $\lambda_\theta$, $\lambda_s$, and $\lambda_C$ to 0.40, 0.30, and 0.30, respectively and $r_{max}$ to 0.55; all values based on those measured from physiology[38]. Correlated Poisson noise, $\eta_{MT}^i$, was then introduced as in ref. [39] to generate variable responses for each trial according to

$$MT_i = f^i(\theta, s, \rho) + \eta_{MT}^i,$$ (12)

where $MT_i$ is the firing rate of the $i$th model MT neuron.

Simulated pursuit initiation responses were generated according to the following steps. Activity from the population of model MT neurons was read off according to two pathways. The first calculated an estimate of the pursuit speed, $\hat{s}$, according to

$$\hat{s}_h = \frac{\sum_{i=1}^N \cos\theta_i' MT_i \log_2 s_i'}{\nu + \sum_{i=1}^N MT_i},$$ (13)

$$\hat{s}_v = \frac{\sum_{i=1}^N \sin\theta_i' MT_i \log_2 s_i'}{\nu + \sum_{i=1}^N MT_i},$$ (14)

and

$$\hat{s} = \sqrt{\hat{s}_h^2 + \hat{s}_v^2},$$ (15)

where $\nu$ was set to 0.05. The second pathway sets the gain according to

$$G_{MT} = \frac{1}{c}\sum_{i=1}^N MT_i \log_2 s_i',$$ (16)

where $c$ was set to make the mean pursuit, across speeds, for the 20 deg patch size equal to 10 deg/s. The simulated pursuit speed, $m$, for each trial was then set to

$$m = (G_{MT} + \eta_G)\hat{s},$$ (17)

where $\eta_G$ was selected each trial from a Gaussian distribution with mean 0 and variance $\sigma^2$. Subsequent analysis of the simulated pursuit responses followed that for the actual pursuit responses.

To compare the covariation between our model neurons and the behavior of the biomimetic circuit to those recorded from MT neurons during pursuit, we selected for those neurons with preferred directions between $-45$ and 45 deg or 135 and 225 deg of the target direction. This matches the differences between target and preferred direction to the distribution studied in the data[24]. We then calculated the trial-by-trial correlations between model neuron output and the output of the entire circuit.

**Reporting summary**. Further information on research design is available in the Nature Research Reporting Summary linked to this article.

## Data availability
All data associated with this study are available at https://doi.org/10.5281/zenodo.5889167[98]. Source data are provided with this paper.

## Code availability
Code for data analysis and implementation of our biomimetic circuit is available at https://doi.org/10.5281/zenodo.5941607[99].

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

## Acknowledgements

This work was supported by NIH grant EY027373 awarded to SGL. We thank Stuart Behling, Timothy R. Darlington, David J. Herzfeld, Nathan J. Hall, Bing Liu, and Leslie C. Osborne for their comments on an earlier version of this manuscript. We also thank Stefanie Tokiyama and Bonnie Bowell for animal assistance.

## Author contributions

S.W.E. and S.G.L. devised the experiment. S.W.E. derived predictions from the simple model. S.W.E. and S.G.L. conceived of the biomimetic model. S.W.E. performed experiments and analyses. S.W.E. and S.G.L. wrote the manuscript.

## Competing interests

The authors declare no competing interests.
