## [Peer Review File · Nature Communications]

Neural structure of a sensory decoder for motor controlREVIEWER COMMENTS

Reviewer #1 (Remarks to the Author):

Egger and Lisberger provide two compelling models that account for the dependence of oculomotor variability on both stimulus speed and stimulus size. The first model is mathematically based; the second is more biologically plausible, based on decoding of a population of neurons with realistic direction and speed tuning, receptive fields and noise correlations. The models themselves are elegant, well explored and potentially provide a highly compelling explanation for the observed data. That said, I do have some concerns about the behavioral methods and data that should be easily addressed by additional analyses and descriptions.

MAJOR

1. The constraints on behavioral task performance do not appear to be particularly strict. The thresholds for rejecting trials (e.g. based on poor tracking or saccades) should be better justified, or analysis of a subset of trials with stricter tracking performed. For example, the initial restriction is that animals must maintain fixation within 3.5deg of the stationary target. Is this a total range of eye movement or a radius from the target? Why was such a large threshold chosen? Regardless, the eye could potentially be 2.5 deg away from the 2deg diameter pursuit target when it appears, which does not seem appropriate.

I am also concerned that eye position during pursuit only needs to be maintained within 3.8-10deg of the target center. Given 900 ms of total movement duration, at 4deg/s that's 3.6deg of movement. But obtaining a reward only requires maintaining the eyes within 3.8deg of target center - even when the target is 2deg diameter, so unless I am mistaken (and I hope I am), this means that animals don't even need to engage in pursuit to obtain a reward! Now, clearly, Fig 1B suggests that the animal is pursuing, so a related concern is that pursuit might only occur on some trials. With a small target, if pursuit was only initiated on a subset of trials, inter-trial variability in speed would be high (as observed) and gain would be low (also as observed). To address this, the authors need to demonstrate that pursuit occurs reliably on every trial. I also suggest that they examine how their results withstand applying a stricter eye-error threshold, so that the reader can be confident that the overall results are not simply a reflection of clever monkeys minimising effort to obtain juice.

2. A related concern is that inter-trial variability in eye speed could simply reflect inter-trial variability in pursuit latency. The most extreme scenario here would have identical eye trajectories on each trial, but with the latency at which the trajectory was initiated being more diverse for small stimuli. To address this, the authors could examine inter-trial variability in latency and demonstrate that it does not trivially account for the gain effect.

3. Smooth eye movements become more reflexive and must be consciously suppressed for large stimuli, whereas they typically require deliberate or intentional initiation when stimuli are small. The authors should discuss how this observation might be related to the size effects reported here. Is there any evidence that animals are engaged in more reflexive behavior for the larger stimuli. This would predict a greater proportion of trials with a small target in which pursuit was not initiated.

MINOR

1. How was the 110-190 ms time window chosen? The 20 ms time window analysis (Line 84 / Supp Fig 1) is reassuring, but also suggests that the Weber fraction 'w_i' begins to converge in later analysis time windows, as pursuit becomes more firmly closed-loop. Does this suggest that the effects described here are predominantly associated with pursuit initiation? What happens if later analysis windows are examined (since pursuit lasted for 750 ms, and the de-saccading was applied out to 250 ms). Can the modeling results account for the convergence of 'w_i' at later time points, maybe based on changes in neural response variability (Fano factor and noise correlations) after stimulus onset?

2. Model comparisons are made between models of varying complexity (e.g. line 81; line 158). That means it is not surprising that the RMSE is smaller for the more complex models, given the additional free parameter(s). It would be helpful to test whether the additional complexity is justified e.g. based on Akaike or Bayesian Information Criterion (AIC/BIC), or an F-test.

3. Figure 1B – why do eye speed traces substantially deviate before 250 ms (4deg/s with 6deg stimulus and all speeds with 20deg stimulus)? Weren't trials with saccades rejected (line 215 Methods)?

4. Between Eq (2) and (3) – The model appears to have been fit to speeds 4/12/20deg/s and tested at 8 & 16 deg/s. Why was this approach taken (e.g. rather than fitting using 80% of data and testing using remaining 10%).

Signed - Nicholas Price

Reviewer #2 (Remarks to the Author):

Happy new year!

Behavioural variability is generally assumed to be inherited from noise in the sensory input prior to sensorimotor transformation and/or injected into motor output after the transformation. The authors find that neither input noise nor output noise can fully account for the pattern of variability found in their experimental data. Instead, they propose that behavioral variability is consistent with a third alternative — computational noise associated with the sensorimotor transformation itself. The idea that behavioral variability can stem from suboptimal computations is not new. However, there have not been very many attempts to identify the latent sources of variability with a view to inform mechanistic models of computation. This study tackles this problem and introduces two key results: (i) Stochasticity in sensorimotor gain contributes to variability in visually-guided smooth-pursuit eye movements, and (ii) A mechanistic model that incorporates known constraints on anatomical connectivity and neural representation can explain key features of both behavioural variability as well as neuron-behaviour correlations in monkey MT cortex. In my view, these findings are significant and have the potential to influence the way we think about how sensorimotor transformations are implemented in neural circuits.

I appreciate the commendable effort put into this work. The writing was quite clear and the data seem to largely support the conclusions. I have several thoughts, suggestions, and questions that are hopefully constructive. My concerns are not minor, but I won't be losing sleep over it either so I encourage the authors to consider them carefully and address them as they see fit.

(1) Lines 86-89: "Intuitively, one might expect that the change in Weber fraction results from integrating the additional motion information associated with larger target sizes. However, ... individual MT neurons are subject to strong noise correlations suggesting limits to the integration of motion information by downstream decoders of MT activity." Also lines 262-263: "the correlated variability of MT neurons appears to limit the increase in information with increasing population size"

Information at the readout is limited primarily by the pattern, not magnitude of noise correlations (e.g. Ruben Moreno-Bote et al. 2014, Bartolo et al. 2020). So the reasoning here is shaky unless the authors can show additional evidence (by decoding a population of simultaneously recorded MT neurons) that information has already saturated within a few hundred neurons that are activated by the smallest target size. Proving this is a tall order unless such data already exist but indirect evidence comes from behavioural studies they cite in lines 96-97: "Weber fractions for speed perception are constant across the range of target sizes used here". In my view, these latter data are enough to argue against the role of sensory noise so there is no need to appeal to the magnitude of noise correlations as it only weakens the argument.

(2) Lines 114-118: “Available physiological evidence rules out the possibility that the change in slope arises from a change in the speed estimate, s^{\wedge} . The response of individual MT neurons to targets of different size has little to no effect on the speed tuning preferences of MT neurons. Therefore, estimates of s^{\wedge} derived by finding the preferred speed at the peak of the population response should not shift as a function of target size, and the increasing slope cannot be explained by an effect of target size on s^{\wedge} .”

The fact that the tuning of individual neurons is unaltered by target size might be necessary but not sufficient to argue that the speed estimate is unaffected by target size. That would depend on how speed is estimated from the activity of MT neurons. The winner-take-all rule is just one specific (and likely suboptimal) way of estimating speed and it is not clear whether it is supported by past findings. I am not completely sure but perhaps reference 49 supports this assertion? If so, it would be helpful to restate the findings of that study to strengthen this argument. On a related note, there is a well documented slow-speed prior expectation that causes speeds to be underestimated more when sensory evidence is weaker (Stocker & Simoncelli 2006). One would expect that increasing the target size should constitute stronger evidence and lead to a smaller bias. So from a purely normative standpoint, the increase in slopes could be simply due to a better estimate of speed for larger target sizes, rather than increase in sensorimotor gain. Reference 44 shows that perceptual variability is unaffected by target size, but are there studies that tested the effect of target size on perceptual bias?

(3) Line 141 above equation (1): “If the sources of noise are statistically independent, then the mean output is... and its variance is...”

The expression for variance has an interaction term $(\sigma^2)(w_s^2)$. If I am not mistaken, this term would only exist if the gain noise (η_G) and sensory noise (η_s) are perfectly correlated such that their covariance $Cov(\eta_G, \eta_s)$ is the product of gain noise variance, σ^2 and sensory noise variance $(w_s^2)(\hat{s}^2)$. So the statistical independence assumption is not valid, no? Alternatively, the authors could modify equation (1) by removing the interaction term -- I don't think this will qualitatively affect any of the results as the inverse dependence on G is still there because of the σ^2 term. Please correct me if I am wrong.

(4) Lines 153-154: “The effective Weber fraction for the model fitted with gain noise agreed with our behavioral observations across monkeys, directions, and target sizes”

This is really cool! However, if I am not mistaken, the model makes yet another prediction. If the behavioural variability is expressed in terms of target speed s instead of mean eye speed μ (Equation 4), then the effective “sensory” weber fraction (which I define as behavioural variance divided by target

speed) should follow the opposite trend — it should increase as the target size increases! No? This is certainly true for the simulations (Figure 5C). But looking at Figure 1C-F, I am not sure whether the data are fully compatible with this prediction. It would be nice if the authors also plotted variance as a function of target speed (in a manner analogous to Figure 1C-F) to address this point. Of course, no model is perfect but this could make way for a useful discussion about the limitations of the model and how to improve it.

(5) Line 159: “Because there is no influence of the motor noise parameter, w_m , on the changes in Weber fraction predicted by the model, we neglected w_m in the analysis for Figure 3C and D (Methods).”

The phrasing is ambiguous and could be misinterpreted. In both models, motor noise affects the weber fraction, just not differently. So better to rephrase to make it clear that you are talking about the difference in model predictions. But a more serious question: Just because the contribution of w_m to the Weber fraction does not differ for the two models, is it okay to neglect them? Including this parameter could improve both models, no? If I am correct, the parameters w_s and w_m are distinguishable because former parameter interacts with σ in equation (1) while the latter does not. So it should be possible to fit w_m . Perhaps it doesn't matter for the conclusion.

(6) Line 319 below equation (4): “Because the term G_i cancels, w_s and σ are the only free parameters for the gain noise hypothesis.”

This cancellation of G_i makes the model-predicted variability independent of G_i . This essentially happens because μ_i is expanded and written as $G_i * s_i$ in the second-term on the right hand side whereas it is left as it is in the first-term. Expanding the first-term will add G_i as an additional free parameter. One can fit G_i to behavioural variability and ask whether these values match the slopes of behavioural mean (Figure 2). It feels like a missed opportunity to not test this because it would validate a much more striking feature of the model that the mean and variability are yoked together by the same sensorimotor gain value.

(7) Lines 326-327: “Based on estimates of the cortical size of the representation as a function of eccentricity in mm² and the density of cortical neurons per mm², we randomly sampled receptive field centers spanning eccentricities of 1 deg to 30 deg.”

Would be useful to know the actual estimates used in the model.

(8) Line 331-332: We set $A = 10$ and $\tau_\theta = 45$. The preferred direction was sampled from a uniform distribution between -90 and 90 . Line 334-335: “We set $A = 10$ and $\tau_s = 1.64$. The preferred speed was sampled from a uniform distribution between 0.5 and 256 deg/s.”

Only directions between -90 and 90 were considered? Maybe this doesn't matter if the extreme angles correspond to target directions. Was the distribution of preferred speeds uniform in log space? Looks like it from Figure 4E.

(9) Equations below line 342 and in Figure 4A: “Activity from the population of model MT neurons was read off according to two pathways.”

“The first calculated an estimate of the pursuit speed according to”... Equation (13)

The left hand side is the estimated speed, \hat{s} whereas the right-hand-side is the center-of-mass of $\log(s_i)$. This tells me that the estimator is severely biased. If neurons are noiseless, the estimated speed is equal to the *logarithm* of the actual speed! Based on my reading of the paper, you never state that the estimator is unbiased but is there a rationale for it to be extremely biased? In fact, larger the speed, greater the bias.

“...where ν was set to 0.05 ”

The other term in the denominator is sum of firing rates of all MT neurons, which I think would be several orders of magnitude larger than ν . So does this term even matter? Perhaps only when all the neurons are silent but was this ever the case?

“The second set the gain according to” ... Equation (14)

The gain is a weighted sum of neural activity where weights are set to be equal to the logarithm of preferred speed, $\log(s_i)$. So looks like neurons that prefer larger speeds contribute disproportionately to the gain value. Again, is there a rationale for this? From reference 30, it appears that in order to predict behaviour, the weighting had to be larger on FEF neurons that were more context-sensitive. Is there a reason to think that MT neurons preferring larger speeds are more context-sensitive? Or am I entirely off the mark here?

“where c was set to make the mean pursuit, across speeds, for the 20 deg patch size equal to the mean across s of $\log(s)$ ”

So on average, the pursuit speed for 20deg target was constrained to be equal to the mean value of $\log(s)$? Was this true in the data?

(10) Line 189: “Gain also increased with target speed due to the speed tuning properties of MT neurons (Figure 4F).”

My sense was that the increase in gain with speed was entirely due to high-speed preferring neurons getting larger weight. If not, better to be explicit about which neuronal tuning property is being referred to here.

(11) Lines 216-218: “We also note that the 2-pathway decoder used in our model (Figure 5A), reproduces the unexpected finding the MT-pursuit correlations were positive whether the target speed was faster or slower than the preferred speed of the MT neuron...”

I believe this finding could be due to the fact that decoding weights of the vast majority of neurons were positive [$= \log_2(s_i)$]? Which is why the model without gain noise also has largely positive correlations? If so, it might be better to rephrase the sentence. The way it reads now, the positive correlation is being attributed entirely to the decoder having two pathways.

(12) Figure 5D-F:

The neuron-behaviour correlations drop to zero for very high target speeds. Is this simply because majority of the neurons no longer respond to the stimulus? If so, it might be better to truncate the x-axis in Figures 5E-F to a smaller value (e.g. 1000%) to avoid confusion. The data in Figure 5D are scant beyond 800% anyway.

(13) Lines 262-264: “the correlated variability of MT neurons appears to limit the increase in information with increasing population size. Indeed, our model’s inclusion of correlated variability in MT neurons with different receptive field locations provides a parsimonious account of results from experiments with both motor and perceptual endpoints.”

This is so only because the model neurons have translation-invariant tuning functions (Ruben Moreno-Bote et al. 2014). Real neural populations likely have some heterogeneity — in this case the information would no longer saturate under the assumed limited-range correlation model (Ecker et al. 2011). So stating this reason weakens the conclusion.

(14) Lines 281-282: “Our findings suggest that the flexibility afforded by independent gain control pathways comes at a cost of increased variability associated with noise that arises through imperfect circuit computations.”

It would be pertinent to discuss how this model relates to the doubly stochastic response model proposed in Goris, Movshon & Simoncelli (2014). In it, the authors used a stochastic gain term to explain super-Poisson variability in individual neurons. The model proposed in this paper is kind of similar in spirit so it would be nice to discuss how they differ.

Other comments:

(1) When reporting results of model comparison, using fraction of variance explained (instead of root-mean-square error) would make the numbers more interpretable.

(2) It would be useful to share the best fit parameter values corresponding to Figure 3C-D somewhere in the paper so the reader can appreciate the relative strengths of sensory noise and gain noise.

(3) The word “decoder” is used throughout the paper when “transformation” would be more appropriate because the gain term transforms the sensory estimate into motor command. Okay to keep the current wording, but it might be better to consider change in a couple of places where replacing makes most sense.

Thanks for the nice study and all the best!

Kaushik Lakshminarasimhan,
Postdoctoral fellow,
Center for theoretical neuroscience,
Zuckerman Institute, Columbia University

Reviewer #3 (Remarks to the Author):

This study examines how the smooth pursuit behavior of macaque monkeys depends on stimulus size, and how that effect might be explained in terms of signal and noise in sensory and non-sensory pathways that mediate the behavior. The behavioral experiments themselves are rather straightforward, and the results are quite clear. The rest of the study involves interpreting this finding, and suggests a pretty radical change to the way that the Lisberger group has previously thought about the noise sources limiting pursuit behavior. While this is interesting, my major problem with the study is that I am not convinced by the arguments that the authors use to discard the pre-existing view.

The main behavioral finding is that the variance of pursuit behavior decreases as the size of a pursuit target (a patch of random dots) increases. The authors argue that this finding “breaks the traditional psychophysical laws of signal-dependent noise” and thus requires a new explanation in terms of the underlying neurobiology. Signal-dependent noise means that variance would grow as the mean of a signal grows. This finding leads the authors to propose that the variability of pursuit behavior is not limited by sensory representations, which was suggested previously by several high-profile papers from the Lisberger lab, but rather has its origins in noise in a gain-control pathway. I have no horse in this game, and I actually find the gain-modulation idea somewhat appealing. However, I was not convinced by the arguments against a sensory origin, and I found a lot of aspects of the proposed neural model questionable. Thus, I am currently unconvinced by the conclusions, but I am certainly open to being swayed if the authors can provide data that can answer my many questions and concerns.

Major concerns:

1) The authors are led to discard the sensory origin idea (made rather famous by previous work from the same laboratory) because they feel that their behavioral data violate the assumption of signal-dependent noise. Specifically, this is because variance of pursuit decreases with stimulus size, and they presume that the signals underlying pursuit behavior would have a mean that increases with stimulus size. This logic would make sense to me if we assume that pursuit is driven by a sensory representation of target velocity (in head coordinates). But it does not make sense to me if I assume that pursuit is driven by a sensory representation of retinal velocity. While the variance of pursuit decreases with target size (Fig. 1 C-F), a critical observation is that the gain of pursuit also increases substantially with

target size (Fig. 2 B,C). Since the gain of pursuit is higher for a large target, the retinal velocity (i.e., retinal slip) of the target is going to be much lower. Thus, from the standpoint of retinal velocity, the reduction of variance for large targets is associated with a reduction of the mean retinal velocity. As a result, the data may still be consistent with signal-dependent noise if one considers the underlying signal to be retinal velocity. As such, it seems critical for the authors to also plot out the variance of pursuit as a function of retinal speed, rather than just plotting it as a function of eye speed. If the data are no longer in violation of signal-dependent noise when considered this way, then it seems to me that the findings would still be consistent with a sensory origin of noise in a representation of retinal velocity (which is what most people think MT carries).

2) A parallel concern pertains to the neural model of MT responses. Here, the authors appear to assume that MT neurons are selective for target velocity rather than retinal velocity, but they don't really state or provide support for this assumption. Since previous studies (i.e., Inaba and Kawano) have reported that MT neurons represent retinal speed instead of head-centered (i.e., target speed), it seemed odd that the authors make this assumption. It would make more sense if they were analyzing responses from time periods before the eyes move (or before eye movements could influence visual responses), but this does not seem to be the case. Since MT neurons begin to respond around 60-70ms, the analysis window of 110-190ms used here leaves plenty of room for MT responses to be influenced by eye pursuit, through both visual and non-visual mechanisms.

3) Following on the above, a key assumption that the authors make about the neural representation is as follows (lines 126-127): "According to everything we know, changing the size of the target should increase the total activity in the MT population response..." Given that MT responses presumably are driven primarily by retinal velocity and that retinal velocity will be lower for the large targets (due to higher pursuit gain), this assumption is far from obvious to me. Clearly, more neurons would be engaged by the larger stimuli. But, counter to that, responses of some neurons will be reduced by surround suppression for larger sizes and responses of almost all MT neurons will be reduced for large target sizes because the retinal speed of the pursuit target will be lower. It's difficult to know what the net effect of these changes will be, and this is one of the reasons why population recordings in MT would have added a lot to constrain the assumptions of this study. If the authors' assumption regarding net MT activity (as stated above) is not rock-solid, then it calls into question the entire inference that the behavior is not consistent with signal-dependent noise in MT. The authors reference Liu et al. (2016) to support their assumptions about how MT population activity changes with size. But in that study, monkeys were not pursuing a target and so retinal speed of the target was not dropping as target size increased.

Specific comments:

4) Lines 17-18: "It is assumed, therefore, that noise is absent from the varied and complex decoding circuits that intervene between sensory representations and motor coordination." I don't agree with this statement. Multiple models that have tried to reconcile perceptual and neural performance have included "pooling noise" or "decision noise". This quantity is effectively the same as noise in the decoding circuitry. So this was a straw man in my view.

5) Lines 79-85: Statistics are clearly called for here to support these quantitative claims.

6) Lines 92-93: "...individual MT neurons are subject to strong noise correlations..." This was a very strange statement since noise correlations can only be defined for populations of two or more neurons. Thus, this was almost nonsensical. Moreover, only differential correlations will limit information, and it is far from clear that correlations would limit information enough to explain the behavior here.

7) In thinking about the neural basis of this behavior, the authors focus entirely on area MT. While I understand that there are many good reasons to consider MT as the primary candidate, especially for typical pursuit target sizes, one has to also consider the possibility that other areas may start to contribute more as target size becomes large. For example, with a 20 deg pursuit target, it may certainly be possible that there is now substantial involvement of the different subdivisions of area MST. If different areas are engaged to different degrees as target size changes, this would also factor into how one expects noise to grow with signal. The authors need to at least acknowledge this possibility and state the caveat that their modelling assumes that MT is the only relevant area, which may not be true.

8) Lines 134-138: The variable $s_{\hat{}}$ here is described as the "sensory estimate", but it is generally unclear in the paper whether the authors assume this to be the sensory estimate of target velocity or retinal velocity. This should be made clear in multiple places where velocity variables are used.

9) Lines 157-160: Again, statistics are needed here.

10) In the "biomimetic" model of MT, the authors model a neural population that has direction and speed tuning curves of fixed widths, and units with identical dynamic ranges. I found this to be a strange choice, since we know that diversity in a population of tuning curves has huge effects on how correlated noise limits information and what types of correlated noise limit information (e.g., Ecker et al 2011; Moreno-Bote et al. 2014). The authors' formulation of the model is precisely the typical oversimplification for which standard limited-range correlations will limit information, whereas this will not be the case with diverse tuning curves. With diverse tuning curves, it will require differential correlations to limit information.

11) Directly following from the above point, it should be noted that the recorded populations were too small to do decoding without having to estimate the covariance matrix. And we know from the work of Pouget's group that estimating the covariance matrix is fraught with difficulties, and likely will misestimate the differential correlations. Given that we also don't know how the differential correlations in MT depend on stimulus size and how they are influenced by extra-retinal signals related to pursuit eye movements (which are known to modulate MT responses), the modelling results seem quite heavily dependent on a set of assumptions that are questionable. Consequently, I am not comfortable accepting the conclusions about sensory origins of noise when the model makes so many assumptions that may be inaccurate.

12) For the above reasons, it really seems important going forward to make population recordings in MT during this behavior to be able to constrain estimates of how the information in the MT population changes with target size, without the need for approximation of correlated noise. I am not saying that the authors need to do this for the manuscript to be reconsidered. But unless I am wrong in my concerns about a lot of the assumptions of the model, I think the authors may have to really scale back and qualify their conclusions considerably.

13) Lines 187-189: Again, this behavior of the neural model seems to be based on the assumption that MT neurons represent target velocity, not retinal velocity.

14) Figure 5F: This result is pitched as providing a good match to the actual data of Fig. 5D. While I certainly agree that it is closer than the model without gain noise, the predicted data of Fig. 5F have a distribution that is clearly quite different from the actual data. So I think the language around this claim needs to be softened up.

15) Lines 281-282: "Our findings suggest that the flexibility afforded by independent gain control pathways comes at a cost of increased variability associated with noise that arises through imperfect circuit computations." Isn't this trivially true? Any time you add a new signal into a computation, noise associated with that new signal is going to combine with noise in the other signals. So unless I am missing something here, this seems like a statement that could have easily been made a priori.

16) Lines 308-311: Here, the authors indicate that they used a much larger fixation window for the 20 deg pursuit targets than for the smaller targets, despite the fact that the pursuit gain for large targets is much closer to unity. This was puzzling and made me wonder about what the monkeys actually do in pursuing these large targets, and why such large fixation windows are needed. It would be useful for the authors to describe the patterns of positional errors that the monkeys make. Since they said something about removing trials with saccades, does this mean that monkeys maintain large positional errors during the pursuit of large targets? Are those positional errors systematic?

17) Lines 326-328: I really did not follow the logic here. 667 MT neurons to represent the central 30 deg of the visual field? Is that supposed to be an estimate for what is actually in the brain? If so, it must be orders of magnitude too small. So the authors should clarify what they intend this statement to mean.

We thank all the reviewers for their thoughtful input. We have resolved all of the major concerns and responded to each other issue.

*We realize that the reviews were quite detailed and our responses are commensurately detailed. Therefore, we start with an **Executive Summary** of the reviewer concerns and a brief statement about how we have resolved them. After, we discuss the issue of differential correlations in detail. Finally, we provided detailed responses each reviewer's comments.*

Executive Summary

Issue #1: Experimental design and data analysis: Each reviewer raised concerns about our experimental design and analysis. We performed control analyses addressing each concern and verify our original results. These details now are included in the text of the paper.

Issue #2: Concerns related to feedback: The reviewers expressed concern about how feedback from eye movements would affect our modeling and results. First, we clarified that we focus on pursuit initiation, which is driven entirely by motion in the period before the eyes begin to move. During this period, image motion and target motion are equal. Second, we justify focusing on initiation as an ideal model system to study decoding. While it would be possible to ask about signal and noise after the initiation of pursuit, feedback limits the interpretability of our data in terms of sensory decoding.

Issue #3: Statistical analyses: The reviewers suggested more in-depth statistical analysis. We did additional tests following their suggestions and are our conclusions did not change.

Issue #4: Differential correlations: The reviewers raised the concern that our model required gain noise due to unrealistic properties in our model MT neurons. The principle concern was whether our use of invariant tuning functions led to the erroneous conclusion that gain noise is required in the decoder. We have revised our model to include a variety of tuning functions and our conclusion that gain noise is necessary still holds. We address the nuanced issue of differential correlations below as well as in the Discussion of the paper.

Issue #5: Questions about our modeling in detail: We have resolved all of them.

Issue #6: Concerns about our detailed model assumptions : The reviewers showed some skepticism in our assumptions regarding the mean and covariance of MT responses with respect to stimulus size. They suggested an extensive MT multi-neuron recording study to validate them. We agree in principle. In practice, the suggested experiment is impossible. It would require the development of new recording technology and 2-3 years of experimentation and data analysis on an unfunded mandate. Given the extensive body of known results from area MT, we ask that we be judged on our model agreement with those results. We assert that our model goes a long way to explaining behavioral variance and MT-behavior correlations based on what is currently known.

Response regarding issue of differential correlations, raised by R2 and R3.

The reviewers bring up the valid concern that our results might be contingent on the unrealistic assumption that tuning curves with respect to speed and direction are invariant.

We followed the reviewers' suggestion and included tuning curves that vary in width and amplitude. Doing so did not change our results.

We now discuss in detail the known theoretical results regarding noise correlations and whether or not they limit information. In the context of our paper and smooth pursuit system in general, there are two ways that caution must be taken applying the results of Moreno-Bote et. al. (2014).

1. The case considered in Moreno-Bote et. al. (2014) applies to a specific covariance structure that is dominated by a single mode. If one extends the results to even one additional mode, then the directions of the correlations and the derivative of the tuning curve no longer have to strictly align to limit information; that is, the structure of the correlations need not be differential to be information limiting (see Supplementary Equation 79 from Moreno-Bote et. al., 2014). The structure of the covariance between model MT neurons used in our results, which is consistent with what is known about MT correlations, has several large modes. In our hands, this correlation structure limits the variance of the decoded output.
2. The claim that "information is limited by the pattern" of noise correlations rests on the assumption that the decoder of the population response is optimal. Suboptimal weighting by linear decoders will limit information even if the correlation structure does not (Moreno-Bote, et. al. 2014). It seems implausible that the brain implements an optimal decoder of MT activity to drive smooth pursuit. (i) It is a non-trivial matter to construct the optimal decoder using a neural circuit. (ii) A linear decoder optimized to decode one stimulus does not generalize to decode other stimuli optimally (Moreno-Bote, et. al. 2014). To achieve optimal decoding across the broad range of stimulus speeds used in our experiment, the linear decoder would have to be adaptive. (iii) Recent experimental results suggest that the decoder used by the brain is aligned with the covariations in a sensory population, not with the optimal decoding direction (Ni et. al. 2018). Our decoder is designed to be consistent with experimental results, not optimal.

Both points highlight the substantial gap between the theoretical results and the conclusions we can make about what limits information in realistic pursuit models. A great deal about the correlation structure of MT and the structure of its downstream decoder remains to be discovered. Because our decoder explains the variance of behavior and MT-behavior correlations while being grounded in known experimental results, we believe we have made an important contribution to closing this gap. We have added a paragraph to the discussion that addresses the results of Moreno-Bote et. al. (2014) and their relation to our simplified claim

that “correlated variability of MT neurons *appears* to limit the increase in information with increasing population size.” (Our stress to point out we have not made a definitive claim).

Responses to Reviewer #1

Egger and Lisberger provide two compelling models that account for the dependence of oculomotor variability on both stimulus speed and stimulus size. The first model is mathematically based; the second is more biologically plausible, based on decoding of a population of neurons with realistic direction and speed tuning, receptive fields and noise correlations. The models themselves are elegant, well explored and potentially provide a highly compelling explanation for the observed data. That said, I do have some concerns about the behavioral methods and data that should be easily addressed by additional analyses and descriptions.

Authors: We appreciate the reviewer’s recognition of our modeling approaches and their potential to explain behavior.

MAJOR

1. The constraints on behavioral task performance do not appear to be particularly strict. The thresholds for rejecting trials (e.g. based on poor tracking or saccades) should be better justified, or analysis of a subset of trials with stricter tracking performed. For example, the initial restriction is that animals must maintain fixation within 3.5deg of the stationary target. Is this a total range of eye movement or a radius from the target? Why was such a large threshold chosen? Regardless, the eye could potentially be 2.5 deg away from the 2deg diameter pursuit target when it appears, which does not seem appropriate.

Authors: First, the facts: we have clarified in the Methods that 3.5 deg is a radius. Second, our reasoning: we chose a large threshold because monkeys perform better when they cannot “feel” the edge of the fixation window. To address the reviewer’s (entirely reasonable) concern, we have reanalyzed our data using only trials where fixation was actually within 2 deg or 2.5 of the target for monkeys R and X, respectively. Then, (1) 78% of the trials had fixation within the respective window of analysis for both monkeys R and X and (2) the results are unchanged when we include only these trials (see Figure R1). We have added a sentence to both the Methods and the Results to include this reassurance.

I am also concerned that eye position during pursuit only needs to be maintained within 3.8-10deg of the target center. Given 900 ms of total movement duration, at 4deg/s that’s 3.6deg of movement. But obtaining a reward only requires maintaining the eyes within 3.8deg of target center - even when the target is 2deg diameter, so unless I am mistaken (and I hope I am), this means that animals don't even need to engage in pursuit to obtain a reward! Now, clearly, Fig 1B suggests that the animal is pursuing, so a related concern is that pursuit might only occur on some trials. With a small target, if pursuit was only initiated on a subset of trials, inter-trial variability in speed would be high (as observed) and gain would be low (also as observed). To address this, the authors need to demonstrate that pursuit occurs reliably on

every trial. I also suggest that they examine how their results withstand applying a stricter eye-error threshold, so that the reader can be confident that the overall results are not simply a reflection of clever monkeys minimising effort to obtain juice.

Authors: Our justification for using large windows during pursuit is that the monkeys have no clue about where in the patch of dots they should fixate. We cannot risk punishing them for perfect behavior even if eye position is not at the center of the patch. We have revisited our data and now state in the text that pursuit speed reached 20% of target speed by 190 ms in 100% of trials for monkey X and 99.5% of trials for monkey R. Inspection of the remaining 0.5% of trials in monkey R revealed that pursuit was still initiated on these trials. In addition, we have superimposed traces for the first 250 ms after target motion onset, including the full analysis interval, and verified visually that a failure to initiate pursuit early in the response is not contaminating the data, also now stated in the text. Unfortunately, the two monkeys used in this study have been euthanized and therefore are not available for control experiments.

2. A related concern is that inter-trial variability in eye speed could simply reflect inter-trial variability in pursuit latency. The most extreme scenario here would have identical eye trajectories on each trial, but with the latency at which the trajectory was initiated being more diverse for small stimuli. To address this, the authors could examine inter-trial variability in latency and demonstrate that it does not trivially account for the gain effect.

Authors: The reviewer is absolutely correct that the variation in latency in response to different size targets can be (and is) different and that these differences can (and do) contribute the observed effects that were attributing to gain noise. To address this, we remove latency variation by realigning the data traces according to objectively-determined single-trial latency and reperformed our analyses. The overall level of variation decreased slightly, but the effect of target size on pursuit variation remained as we originally reported it. The paper now reports only the results for the data after removing the variation in latency.

3. Smooth eye movements become more reflexive and must be consciously suppressed for large stimuli, whereas they typically require deliberate or intentional initiation when stimuli are small. The authors should discuss how this observation might be related to the size effects reported here. Is there any evidence that animals are engaged in more reflexive behavior for the larger stimuli. This would predict a greater proportion of trials with a small target in which pursuit was not initiated.

Authors: Again, pursuit was initiated in all the trials we consider in our analyses. The question of whether the smooth eye movement response is more reflexive for bigger targets is unanswerable. We assume that the monkey's work status is equal across target sizes because: (1) the target sizes are interleaved randomly so that the monkey must be in the same state before each presentation; (2) the responses are probably all more or less reflexive because the only target in the field is moving and the monkey has been trained to be a tracking machine; and (3) we are working with highly-trained and fluid-motivated monkeys whose mindset is probably "follow that target, it leads to fluid". The literature has identified a "reflexive" smooth

eye tracking response that is called ocular following, but it has so many parallels to the pursuit we study that it is difficult to consider them as fully separate. (1) Ocular following uses the same basic anatomical pathway as pursuit. (2) Ocular following is subject to gain modulation with enhancement of visual-motor gain following saccades, just like pursuit. (3) Actually, the response is generally considered to be “reflexive” ocular following only for much larger stimuli than we used. We now address this issue briefly in the Discussion.

MINOR

1. How was the 110-190 ms time window chosen? The 20 ms time window analysis (Line 84 / Supp Fig 1) is reassuring, but also suggests that the Weber fraction ‘ w_i ’ begins to converge in later analysis time windows, as pursuit becomes more firmly closed-loop. Does this suggest that the effects described here are predominantly associated with pursuit initiation? What happens if later analysis windows are examined (since pursuit lasted for 750 ms, and the de-saccading was applied out to 250 ms). Can the modeling results account for the convergence of ‘ w_i ’ at later time points, maybe based on changes in neural response variability (Fano factor and noise correlations) after stimulus onset?

Authors: First, the facts: the window of 110-190 ms was chosen based on previous research indicating that motor activity in this window is driven entirely by the feed-forward stimulus (e.g. open-loop pursuit; Lisberger and Westbrook, 1985). Averaging across the entire window ensured robust estimates of variation, although we see the same effects in 20 ms intervals (Supplementary Figure 1). The more granular estimates of the Weber fractions became more stable after removing variation due to trial-by-trial differences in latency (as suggested by the reviewer above), compared to the previous version of Supplementary Figure 1.

Second, the bigger question of later analysis windows: data on Fano factor and noise correlations are available only for passive viewing conditions, which are equivalent to the open-loop interval we analyze. Beyond that, feedback will (1) change the visual input and responses; (2) eventually reduce variation as an unavoidable consequence of closed-loop control; and (3) create trial by trial variation that is due to variation in the initial response and is out of our control. The analysis at this stage would be ill-posed. We note these points at the end of the first paragraph of the results.

Third, we think it is outside the scope of this paper to consider models that fully reflect the temporal dynamics of MT, FEF, cerebellar, brainstem, and pursuit responses. At the moment, we do not have a firm model that relates the time course of MT responses to either speed estimation over time or how this ought to impact responses in FEFsem.

2. Model comparisons are made between models of varying complexity (e.g. line 81; line 158). That means it is not surprising that the RMSE is smaller for the more complex models, given the additional free parameter(s). It would be helpful to test whether the additional complexity is justified e.g. based on Akaike or Bayesian Information Criterion (AIC/BIC), or an F-test.

Authors: We appreciate the reviewer's concern about the benefits of additional parameters. We note, however, that we performed comparisons based on left out *conditions* (model fit to 4/12/20 deg/s and tested on 8/16 deg/s) to eliminate overfitting advantages of increased model complexity. Now, we also performed a bootstrap analysis as follows: we fit the models in the paper to 50% of the data (across trials) and found the RMSE of each model on the variance computed from the remaining 50% of trials. We repeat this analysis 1000 times, randomly sampling fit/test trial on each iteration and used the distribution of RMSEs to determine the significance of the fit improvement. We include the new statistics in the Results.

3. Figure 1B – why do eye speed traces substantially deviate before 250 ms (4deg/s with 6deg stimulus and all speeds with 20deg stimulus)? Weren't trials with saccades rejected (line 215 Methods)?

Authors: This is our error. Trials with the *midpoint* of saccades before 270 ms were rejected. In our analysis we assume a 40 ms saccade duration, which matches typical catch-up saccade amplitudes (< 5 deg). However, monkeys on occasion made larger saccades in response to the larger targets as they entered the closed-loop period of pursuit, a consequence of using large target windows, as discussed above. Therefore, the early portion of some of these saccades leaked into our averaging window. Because the window for quantitative analysis ended 190 ms following motion onset, these saccades do not affect our results. To avoid misleading readers, we revised Figure 1B so that mean pursuit from 0-210 ms is plotted.

4. Between Eq (2) and (3) – The model appears to have been fit to speeds 4/12/20deg/s and tested at 8 & 16 deg/s. Why was this approach taken (e.g. rather than fitting using 80% of data and testing using remaining 10%).

Authors: The advantage of our approach is that it maximized the number of trials used to estimate the variance for each condition. Following the reviewer's advice, we have verified that we obtain the same answer if we use a random sample of 50% of trials and then test model fits to variance calculated from the remaining 50% (see above). 20% of trials was insufficient to reliably estimate variance.

Responses to Reviewer #2

Happy new year!

Authors: To you as well! We thank you for taking the time during this busy season to review our work.

Behavioural variability is generally assumed to be inherited from noise in the sensory input prior to sensorimotor transformation and/or injected into motor output after the transformation. The authors find that neither input noise nor output noise can fully account for the pattern of variability found in their experimental data. Instead, they propose that behavioral variability is consistent with a third alternative — computational noise associated

with the sensorimotor transformation itself. The idea that behavioral variability can stem from suboptimal computations is not new. However, there have not been very many attempts to identify the latent sources of variability with a view to inform mechanistic models of computation. This study tackles this problem and introduces two key results: (i) Stochasticity in sensorimotor gain contributes to variability in visually-guided smooth-pursuit eye movements, and (ii) A mechanistic model that incorporates known constraints on anatomical connectivity and neural representation can explain key features of both behavioural variability as well as neuron-behaviour correlations in monkey MT cortex. In my view, these findings are significant and have the potential to influence the way we think about how sensorimotor transformations are implemented in neural circuits.

I appreciate the commendable effort put into this work. The writing was quite clear and the data seem to largely support the conclusions. I have several thoughts, suggestions, and questions that are hopefully constructive. My concerns are not minor, but I won't be losing sleep over it either so I encourage the authors to consider them carefully and address them as they see fit.

Authors: We are glad that the reviewer recognizes how our work fills a gap in our current understanding of signal and noise in sensory-motor behavior – specifically that the noise involved in a specific computational pathway can give rise to a specific signature in behavioral variation.

(1) Lines 86-89: “Intuitively, one might expect that the change in Weber fraction results from integrating the additional motion information associated with larger target sizes. However, ... individual MT neurons are subject to strong noise correlations suggesting limits to the integration of motion information by downstream decoders of MT activity.” Also lines 262-263: “the correlated variability of MT neurons appears to limit the increase in information with increasing population size”

Information at the readout is limited primarily by the pattern, not magnitude of noise correlations (e.g. Ruben Moreno-Bote et al. 2014, Bartolo et al. 2020). So the reasoning here is shaky unless the authors can show additional evidence (by decoding a population of simultaneously recorded MT neurons) that information has already saturated within a few hundred neurons that are activated by the smallest target size. Proving this is a tall order unless such data already exist but indirect evidence comes from behavioural studies they cite in lines 96-97: “Weber fractions for speed perception are constant across the range of target sizes used here”. In my view, these latter data are enough to argue against the role of sensory noise so there is no need to appeal to the magnitude of noise correlations as it only weakens the argument.

Authors: See discussion before Responses to Reviewer #1.

(2) Lines 114-118: “Available physiological evidence rules out the possibility that the change in slope arises from a change in the speed estimate, s^{\wedge} . The response of individual MT neurons to

targets of different size has little to no effect on the speed tuning preferences of MT neurons. Therefore, estimates of s^{\wedge} derived by finding the preferred speed at the peak of the population response should not shift as a function of target size, and the increasing slope cannot be explained by an effect of target size on s^{\wedge} ."

The fact that the tuning of individual neurons is unaltered by target size might be necessary but not sufficient to argue that the speed estimate is unaffected by target size. That would depend on how speed is estimated from the activity of MT neurons. The winner-take-all rule is just one specific (and likely suboptimal) way of estimating speed and it is not clear whether it is supported by past findings. I am not completely sure but perhaps reference 49 supports this assertion? If so, it would be helpful to restate the findings of that study to strengthen this argument.

Authors: Our reference to the "peak of the population response" was intended to provide an intuitive description of how decoder of speed works in general. The literature supports a vector averaging algorithm similar to the used by us in the current paper (e.g. Lisberger and Ferrera, 1997; Priebe et. al., 2001). To make this clear, we have revised lines 114-118.

On a related note, there is a well documented slow-speed prior expectation that causes speeds to be underestimated more when sensory evidence is weaker (Stocker & Simoncelli 2006). One would expect that increasing the target size should constitute stronger evidence and lead to a smaller bias. So from a purely normative standpoint, the increase in slopes could be simply due to a better estimate of speed for larger target sizes, rather than increase in sensorimotor gain. Reference 44 shows that perceptual variability is unaffected by target size, but are there studies that tested the effect of target size on perceptual bias?

Authors: The reviewer's observation about potential effects of priors on pursuit behavior is astute and, in fact, a major finding in previous work from our lab. Pursuit initiation responses do reflect the relative weighting of prior and sensory evidence, as suggested by the reviewer. The bias is toward the low speeds, as in perception, and is adaptable in relation to the mean of the distribution of speeds used in the task when considering the behavior of highly trained monkeys (Darlington, et. al. 2017). Our model, and the data and modeling of Darlington et al. (2017, 2018) show that a reliability-weighted combination of the prior and the sensory data is implemented by the control of sensory-motor gain and the point of our model is that the same computational mechanism also can implement the "more reliable" motion signals that arise from larger targets. We have written explicitly about this in the Discussion and hope that it is now clearer for the reader. We note that the idea of a slow speed prior pre-dates Stocker and Simoncelli (2006) and was formalized by Weiss et. al. (2002), which we cite accordingly.

(3) Line 141 above equation (1): "If the sources of noise are statistically independent, then the mean output is... and its variance is..."

The expression for variance has an interaction term $(\sigma^2)*(w_s^2)$. If I am not mistaken, this term would only exist if the gain noise (η_G) and sensory noise (η_s) are perfectly

correlated such that their covariance $\text{Cov}(\eta_G, \eta_s)$ is the product of gain noise variance, σ^2 and sensory noise variance $(w_s^2)(\hat{s}^2)$. So the statistical independence assumption is not valid, no? Alternatively, the authors could modify equation (1) by removing the interaction term -- I don't think this will qualitatively affect any of the results as the inverse dependence on G is still there because of the σ^2 term. Please correct me if I am wrong.

Authors: Despite the independence of the noise sources, the term $(\sigma^2)(w_s^2)$ does, in fact, remain. The important thing to note is that the related stochastic terms, η_G and η_s , are squared in the definition of the variance. All terms where both stochastic variables are squared when they appear together do not average to zero and appear in the final definition of the variance. We have included the derivation of the mean and variance of the gain noise model in the Supplementary Materials as Appendix A.1. In practice, because both σ and w_s are less than one, the term $\sigma^2 w_s^2$ has little effect on the results.

(4) Lines 153-154: "The effective Weber fraction for the model fitted with gain noise agreed with our behavioral observations across monkeys, directions, and target sizes"

This is really cool! However, if I am not mistaken, the model makes yet another prediction. If the behavioural variability is expressed in terms of target speed s instead of mean eye speed μ (Equation 4), then the effective "sensory" weber fraction (which I define as behavioural variance divided by target speed) should follow the opposite trend — it should increase as the target size increases! No? This is certainly true for the simulations (Figure 5C). But looking at Figure 1C-F, I am not sure whether the data are fully compatible with this prediction. It would be nice if the authors also plotted variance as a function of target speed (in a manner analogous to Figure 1C-F) to address this point. Of course, no model is perfect but this could make way for a useful discussion about the limitations of the model and how to improve it.

Authors: It is important to note that our model predictions would be for sensory *estimates*, not the sensory input itself. The predictions of the reviewer would only be strictly true in the condition where the estimator is linear (see Appendix A.2), which is not what we observed in the behavior. Further, because w_s and G are both less than zero, $w_s^2 G^2$ is very small. As a result, the gain's contribution to the 'sensory' Weber fraction will be small and variance across size conditions will be similar when expressed as a function of target speed. Expressing variance as a function of the mean output (as is standard in the field), makes the differences apparent and experimentally testable.

(5) Line 159: "Because there is no influence of the motor noise parameter, w_m , on the changes in Weber fraction predicted by the model, we neglected w_m in the analysis for Figure 3C and D (Methods)."

The phrasing is ambiguous and could be misinterpreted. In both models, motor noise affects the weber fraction, just not differently. So better to rephrase to make it clear that you are talking about the difference in model predictions. But a more serious question: Just because the contribution of w_m to the Weber fraction does not differ for the two models, is it okay to

neglect them? Including this parameter could improve both models, no? If I am correct, the parameters w_s and w_m are distinguishable because former parameter interacts with σ in equation (1) while the latter does not. So it should be possible to fit w_m . Perhaps it doesn't matter for the conclusion.

Authors: The reviewer is correct that, in principle, the interaction between w_s and σ in equation (1) suggests that the impact of w_s and w_m could be distinguished. In practice, because both w_s and σ are small, their interaction is negligible and the impact of w_m and w_s is nearly identical. We have fit the data with a model that includes w_m , but there is no difference in the results (included in Supplementary Table 1). We thank the reviewer for pointing out our ambiguous language and we have removed this sentence from the main results and included a brief statement summarizing the above in the Methods.

(6) Line 319 below equation (4): "Because the term G_i cancels, w_s and σ are the only free parameters for the gain noise hypothesis."

This cancellation of G_i makes the model-predicted variability independent of G_i . This essentially happens because μ_i is expanded and written as $G_i * s_i$ in the second-term on the right hand side whereas it is left as it is in the first-term. Expanding the first-term will add G_i as an additional free parameter. One can fit G_i to behavioural variability and ask whether these values match the slopes of behavioural mean (Figure 2). It feels like a missed opportunity to not test this because it would validate a much more striking feature of the model that the mean and variability are yoked together by the same sensorimotor gain value.

Authors: As above, it is important to note that the gain noise model makes predictions based on sensory estimates and not the sensory input itself, so the gain fit to the mean and variance of responses would only be expected to equal the slope of the regression function in the limited case where sensory estimation is linear, which does not appear to be the case for our data. Still, following the reviewer's suggestion, we fit the gain noise model with a free parameter for the gain in each condition and compared these to those which we found via regression. As shown in Figure R2, there is a strong positive correlation between the gain estimated in different ways, although the relationship is not perfect. We have also included these results as Supplementary Figure 3.

(7) Lines 326-327: "Based on estimates of the cortical size of the representation as a function of eccentricity in mm² and the density of cortical neurons per mm², we randomly sampled receptive field centers spanning eccentricities of 1 deg to 30 deg."

Would be useful to know the actual estimates used in the model.

Authors: We have modified lines 326-327 to indicate the exact probability density function we used. We previously chose a density of 20 neurons per mm, but this number was admittedly arbitrary. In our revised draft we base the number of model neurons used in the main text (now 1280) on the balance between computational demands and the effect of neuron number on

the change in Weber fractions, total variance, and the magnitude of MT-behavior correlations. We now say so in the revised methods.

(8) Line 331-332: We set $A = 10$ and $\tau\theta = 45$. The preferred direction was sampled from a uniform distribution between -90 and 90 . Line 334-335: “We set $A = 10$ and $\tau s = 1.64$. The preferred speed was sampled from a uniform distribution between 0.5 and 256 deg/s.”

Only directions between -90 and 90 were considered? Maybe this doesn't matter if the extreme angles correspond to target directions. Was the distribution of preferred speeds uniform in log space? Looks like it from Figure 4E.

Authors: Preferred speeds were uniformly distributed in log space. Preferred directions were sampled from -90 to 90 deg relative to the direction of motion. Sampling from -180 to 180 did not alter the results because our model neurons were unresponsive for null-direction motion. In response to this and other reviewer comments, we revised figures in the text to include neurons with preferred directions between -180 and 180 , variable speed and direction tuning widths, and variable amplitudes.

(9) Equations below line 342 and in Figure 4A: “Activity from the population of model MT neurons was read off according to two pathways.”

“The first calculated an estimate of the pursuit speed according to”... Equation (13)

The left hand side is the estimated speed, \hat{s} whereas the right-hand-side is the center-of-mass of $\log(s_i)$. This tells me that the estimator is severely biased. If neurons are noiseless, the estimated speed is equal to the *logarithm* of the actual speed! Based on my reading of the paper, you never state that the estimator is unbiased but is there a rationale for it to be extremely biased? In fact, larger the speed, greater the bias.

Authors: As the reviewer correctly points out, the ‘speed estimation’ pathway does, in fact, estimate $\log_2(\text{speed})$, which would be a poor estimator if it were not compensated for by the gain pathway. It is important to remember that gain increases with speed as well, and the final output of the system approximately reproduces the biases we observe in actual behavior. We now comment on this in the text.

“...where ν was set to 0.05 ”

The other term in the denominator is sum of firing rates of all MT neurons, which I think would be several orders of magnitude larger than ν . So does this term even matter? Perhaps only when all the neurons are silent but was this ever the case?

Authors: As pointed out by the reviewer, the parameter ν is inconsequential in practice for our model as we do not present a case where all MT neurons are silent. We chose this form to

be consistent with previous center of mass (i.e. vector average) models of decoding MT (e.g. Priebe and Lisberger, 2004; Huang and Lisberger 2009).

“The second set the gain according to” ... Equation (14)

The gain is a weighted sum of neural activity where weights are set to be equal to the logarithm of preferred speed, $\log(s_i)$. So looks like neurons that prefer larger speeds contribute disproportionately to the gain value. Again, is there a rationale for this? From reference 30, it appears that in order to predict behaviour, the weighting had to be larger on FEF neurons that were more context-sensitive. Is there a reason to think that MT neurons preferring larger speeds are more context-sensitive? Or am I entirely off the mark here?

Authors: We chose this weighting based on previous results that have demonstrated that pursuit gain increases with the speed of the target (Schwartz and Lisberger, 1994). As FEFsem neurons are not explicitly modeled here, it is difficult to address the detail mentioned by the reviewer. But we do not think that MT neurons are context-sensitive at all – it is the FEFsem neurons that have been demonstrated to be context sensitive in a way that adjusts gain. In response to the reviewer’s specific question about the choice of the weight vector, we think they are taking the details of our model too literally and we agree that the data we are modeling do not really constrain the detailed shape of the weight vector. It seemed distracting to go into this in the paper, even in the Methods.

“where c was set to make the mean pursuit, across speeds, for the 20 deg patch size equal to the mean across s of $\log(s)$ ”

So on average, the pursuit speed for 20deg target was constrained to be equal to the mean value of $\log(s)$? Was this true in the data?

Authors: The normalization reported in the initial methods was in error. The normalization is set for the revised model to make the response to the 20 deg target 10 deg/s on average (across trials and target speeds). This was chosen to approximate what we observed in monkey behavior.

(10) Line 189: “Gain also increased with target speed due to the speed tuning properties of MT neurons (Figure 4F).”

My sense was that the increase in gain with speed was entirely due to high-speed preferring neurons getting larger weight. If not, better to be explicit about which neuronal tuning property is being referred to here.

Authors: The increase in gain with target speed is related both to the log-normal shape of the speed tuning curve and the weighting profile given to MT neurons in our model. We have clarified in the text.

(11) Lines 216-218: “We also note that the 2-pathway decoder used in our model (Figure 5A), reproduces the unexpected finding the MT-pursuit correlations were positive whether the target speed was faster or slower than the preferred speed of the MT neuron...”

I believe this finding could be due to the fact that decoding weights of the vast majority of neurons were positive $[= \log_2(s_i)]$? Which is why the model without gain noise also has largely positive correlations? If so, it might be better to rephrase the sentence. The way it reads now, the positive correlation is being attributed entirely to the decoder having two pathways.

Authors: We have revised the text to complete our thought. A “decoder” that uses vector averaging to estimate target speed predicts that the MT-pursuit correlations will be negative for neurons with preferred speed less than target speed because increases in their firing will pull the center-of-mass of the population response towards lower speeds. The data disagree with this prediction and the two-pathway model reproduces the data much better because the gain pathway results in a uniform relationship between MT-pursuit correlations and target speed relative to preferred speed.

In regard to the decoding computation, we have corrected an error in the original version of the paper. In the original, we estimated speed using an opponent version of equation 13 and applied the gain as in equations 14 and 15, but then applied a nonlinearity ($2^{(m^1/4)}$), with m equal to the output of equation 15. In our revised submission, we have omitted the nonlinearity and correctly reported the opponent strategy in the estimation pathway (see equations 13-15 in the revised manuscript). The results do not critically depend on these details, but our revised model better matches the overall variance and the distribution of MT-behavior correlations observed. We apologize for this error in reporting our methods. It does not affect the conclusions.

(12) Figure 5D-F: The neuron-behaviour correlations drop to zero for very high target speeds. Is this simply because majority of the neurons no longer respond to the stimulus? If so, it might be better to truncate the x-axis in Figures 5E-F to a smaller value (e.g. 1000%) to avoid confusion. The data in Figure 5D are scant beyond 800% anyway.

Authors: We now plot the results from the revised model with identical axes and for data limited to be between 25 and 1000% of preferred speed in the revised Figure 5 following the reviewer suggestion.

(13) Lines 262-264: “the correlated variability of MT neurons appears to limit the increase in information with increasing population size. Indeed, our model’s inclusion of correlated variability in MT neurons with different receptive field locations provides a parsimonious account of results from experiments with both motor and perceptual endpoints.”

This is so only because the model neurons have translation-invariant tuning functions (Ruben Moreno-Bote et al. 2014). Real neural populations likely have some heterogeneity — in this

case the information would no longer saturate under the assumed limited-range correlation model (Ecker et al. 2011). So stating this reason weakens the conclusion.

Authors: We have now included simulations with heterogenous tuning curves. The key results do not change. It is important to note that the results of Ecker et. al. (2011) make the simplifying assumption that limited range correlations only exist along the stimulus to be decoded (e.g. speed), whereas our model (and populations of real MT neurons) have noise correlations along several other key dimensions (e.g. direction, receptive field position). As discussed above, the added complexity of the covariance matrix make it more likely that information will be limited (Moreno-Bote et. al. 2014). As we originally pointed out, including realistic and complex noise correlations in our model provides a parsimonious account of our results and those from perceptual psychophysics that observe no change in Weber fractions with increasing stimulus size (de Bruyn and Orbanm 1988; Vergheze and Stone, 1995).

(14) Lines 281-282: “Our findings suggest that the flexibility afforded by independent gain control pathways comes at a cost of increased variability associated with noise that arises through imperfect circuit computations.”

It would be pertinent to discuss how this model relates to the doubly stochastic response model proposed in Goris, Movshon & Simoncelli (2014). In it, the authors used a stochastic gain term to explain super-Poisson variability in individual neurons. The model proposed in this paper is kind of similar in spirit so it would be nice to discuss how they differ.

Authors: We have now included a brief discussion of Goris et. al.

Other comments:

(1) When reporting results of model comparison, using fraction of variance explained (instead of root-mean-square error) would make the numbers more interpretable.

Authors: We now include variance explained in Supplementary Table 1. We have kept the use of RMSE in the main text because variance explained is a particularly bad measure of models that scale linearly with each other, such as models of signal dependent noise used in our paper.

(2) It would be useful to share the best fit parameter values corresponding to Figure 3C-D somewhere in the paper so the reader can appreciate the relative strengths of sensory noise and gain noise.

Authors: The best fit parameters are now included in Supplementary Table 1.

(3) The word “decoder” is used throughout the paper when “transformation” would be more

appropriate because the gain term transforms the sensory estimate into motor command. Okay to keep the current wording, but it might be better to consider change in a couple of places where replacing makes most sense.

Authors: We really meant decoder. One of the points of our paper is that we define the decoder conceptually as the transformation between sensory representation and behavior. So, we need to evaluate the decoder as the whole neural circuit that is downstream from the sensory representation. We are trying to create a paradigm shift in terms of how people in the field think about decoders – less as equations to be optimized and more as complex, parallel components in realistic neural circuits. We have revised the text in the introduction and discussion to more clearly define a decoder as the complete set of operations that transformation sensory representations to motor output.

Thanks for the nice study and all the best!

Responses to Reviewer #3

This study examines how the smooth pursuit behavior of macaque monkeys depends on stimulus size, and how that effect might be explained in terms of signal and noise in sensory and non-sensory pathways that mediate the behavior. The behavioral experiments themselves are rather straightforward, and the results are quite clear. The rest of the study involves interpreting this finding, and suggests a pretty radical change to the way that the Lisberger group has previously thought about the noise sources limiting pursuit behavior. While this is interesting, my major problem with the study is that I am not convinced by the arguments that the authors use to discard the pre-existing view.

The main behavioral finding is that the variance of pursuit behavior decreases as the size of a pursuit target (a patch of random dots) increases. The authors argue that this finding “breaks the traditional psychophysical laws of signal-dependent noise” and thus requires a new explanation in terms of the underlying neurobiology. Signal-dependent noise means that variance would grow as the mean of a signal grows. This finding leads the authors to propose that the variability of pursuit behavior is not limited by sensory representations, which was suggested previously by several high-profile papers from the Lisberger lab, but rather has its origins in noise in a gain-control pathway. I have no horse in this game, and I actually find the gain-modulation idea somewhat appealing. However, I was not convinced by the arguments against a sensory origin, and I found a lot of aspects of the proposed neural model questionable. Thus, I am currently unconvinced by the conclusions, but I am certainly open to being swayed if the authors can provide data that can answer my many questions and concerns.

Authors: We appreciate the reviewer’s skepticism in light of conclusion from the lab for a primarily sensory origin of pursuit variance. We address the specific concerns below. But two general points are worth making here. (1) The experiments of Osborne et. al. used a limited

range of pursuit stimuli and contexts creating a situation where noise from the gain system would be indistinguishable from sensory noise. For gain noise to be revealed, one needs to design an experiment that systematically changes the mean gain, something which was intentionally avoided in earlier experiments. (2) One of the important observations in the present paper is that the degree of noise and correlation in the MT population response does not allow a reasonable model to reproduce either the variance of the behavioral output or the recorded “MT-pursuit correlations”. Adding either noise or correlation in the MT population response worsens the disagreement between the model’s and the monkey’s MT-pursuit correlations. Downstream noise is definitely required. One advantage of gain noise is that under the conditions used by Osborne et al, gain noise is indistinguishable from sensory noise, meaning that their findings and conclusions remain valid, simply incomplete. The other advantage of gain noise is that it accounts for the effects of target size on pursuit initiation. We have addressed this key issue more explicitly in the revised text.

Major concerns:

1) The authors are led to discard the sensory origin idea (made rather famous by previous work from the same laboratory) because they feel that their behavioral data violate the assumption of signal-dependent noise. Specifically, this is because variance of pursuit decreases with stimulus size, and they presume that the signals underlying pursuit behavior would have a mean that increases with stimulus size. This logic would make sense to me if we assume that pursuit is driven by a sensory representation of target velocity (in head coordinates). But it does not make sense to me if I assume that pursuit is driven by a sensory representation of retinal velocity. While the variance of pursuit decreases with target size (Fig. 1 C-F), a critical observation is that the gain of pursuit also increases substantially with target size (Fig. 2 B,C). Since the gain of pursuit is higher for a large target, the retinal velocity (i.e., retinal slip) of the target is going to be much lower.

Thus, from the standpoint of retinal velocity, the reduction of variance for large targets is associated with a reduction of the mean retinal velocity. As a result, the data may still be consistent with signal-dependent noise if one considers the underlying signal to be retinal velocity. As such, it seems critical for the authors to also plot out the variance of pursuit as a function of retinal speed, rather than just plotting it as a function of eye speed. If the data are no longer in violation of signal-dependent noise when considered this way, then it seems to me that the findings would still be consistent with a sensory origin of noise in a representation of retinal velocity (which is what most people think MT carries).

Authors: The reviewer has misunderstood our experiment and analysis. We analyzed only the first 80 ms of pursuit, which is driven by the visual signals that arise from target motion before the eyes have started to move: target velocity in head coordinates is exactly equal to target velocity in retinal coordinates (aka image velocity). We think this obviates the reviewer’s concern, which would be entirely legitimate if we had conflated target velocity with image velocity. We added a sentence to the text to explain this situation in a way that we hope will prevent other readers for reaching the same misunderstanding.

2) A parallel concern pertains to the neural model of MT responses. Here, the authors appear to assume that MT neurons are selective for target velocity rather than retinal velocity, but they don't really state or provide support for this assumption. Since previous studies (i.e., Inaba and Kawano) have reported that MT neurons represent retinal speed instead of head-centered (i.e., target speed), it seemed odd that the authors make this assumption. It would make more sense if they were analyzing responses from time periods before the eyes move (or before eye movements could influence visual responses), but this does not seem to be the case. Since MT neurons begin to respond around 60-70ms, the analysis window of 110-190ms used here leaves plenty of room for MT responses to be influenced by eye pursuit, through both visual and non-visual mechanisms.

Authors: Here, the reviewer's question is about the exact timing of feedback and its effect on the population response in MT. This is a legitimate question, and one that is fully answered by Figures 9 and 10 in Lisberger and Westbrook (1985). They showed that the presence or absence of feedback does not alter the eye movements in the first 100 ms of pursuit and, indeed, the effect of feedback is paradoxically late so that the first 120 ms or more is part of the open-loop interval. So, we would argue that the reviewer's concern does not have an impact in practice. We clarify in the text that our model neurons are meant to reflect the response of MT neurons over the first 80 ms of target motion, before eye movements change image velocity.

3) Following on the above, a key assumption that the authors make about the neural representation is as follows (lines 126-127): "According to everything we know, changing the size of the target should increase the total activity in the MT population response..." Given that MT responses presumably are driven primarily by retinal velocity and that retinal velocity will be lower for the large targets (due to higher pursuit gain), this assumption is far from obvious to me. Clearly, more neurons would be engaged by the larger stimuli. But, counter to that, responses of some neurons will be reduced by surround suppression for larger sizes and responses of almost all MT neurons will be reduced for large target sizes because the retinal speed of the pursuit target will be lower. It's difficult to know what the net effect of these changes will be, and this is one of the reasons why population recordings in MT would have added a lot to constrain the assumptions of this study. If the authors' assumption regarding net MT activity (as stated above) is not rock-solid, then it calls into question the entire inference that the behavior is not consistent with signal-dependent noise in MT. The authors reference Liu et al. (2016) to support their assumptions about how MT population activity changes with size. But in that study, monkeys were not pursuing a target and so retinal speed of the target was not dropping as target size increased.

Authors: With all due respect, we think that the reviewer has set the bar unfairly high here. (1) They continue to be concerned about whether MT population responses are driven by target motion or image motion, something we have addressed above by pointing out that the behavior we measured is truly in the open-loop interval, demonstrably before feedback affects the behavioral response. (2) We agree that the details of the model of MT responses is potentially important, but we also point out that we have taken great care to explain our assumptions and to verify that they don't affect the main conclusions about model behavior. In

particular, we systematically varied the degree to which our model MT neurons were subject to surround suppression and showed that surround suppression was not an important factor in determining the variance of decoder output as a function of target size (previously Supplementary Figure 3, now Supplementary Figure 4). (3) Given that there is abundant MT data in the literature on which to base our model MT population response, we argue that it is excessive to require us to conduct a 2-3 year research study to obtain data that would not improve on what is already known. It is true that most of our knowledge of the parametric variation of MT responses is based on experiments during passive stimulation rather than during pursuit. However, because we are studying the open-loop response, the available data from passive viewing are more than adequate to make a good model.

Specific comments:

4) Lines 17-18: “It is assumed, therefore, that noise is absent from the varied and complex decoding circuits that intervene between sensory representations and motor coordination.” I don’t agree with this statement. Multiple models that have tried to reconcile perceptual and neural performance have included “pooling noise” or “decision noise”. This quantity is effectively the same as noise in the decoding circuitry. So this was a straw man in my view.

Authors: We originally discussed this issue in the following paragraph, but agree that this approach led us to misrepresent our argument. More precisely, we meant that the *signal dependence* of noise in the sensorimotor literature has assumed that noise is either sensory or motor in origin. Importantly, pooling/decision noise, as implemented previously (e.g. Shadlen et. al. 1996), would result in $VAR = (w_s^2 + w_m^2)\mu^2 + \sigma_{pooling}^2$. Note that the term related to pooling noise is not dependent on gain and does not increase with μ . Therefore, pooling/decision noise is not predicted to cause the shifts in signal dependent noise that we observe in behavior. We have revised the introduction to more precisely state our argument.

5) Lines 79-85: Statistics are clearly called for here to support these quantitative claims.

Authors: As discussed in response to reviewer 1’s comments, we have included a bootstrap analysis to determine the significance of the fit improvement. The statistics validated our previous conclusions.

6) Lines 92-93: “...individual MT neurons are subject to strong noise correlations...” This was a very strange statement since noise correlations can only be defined for populations of two or more neurons. Thus, this was almost nonsensical. Moreover, only differential correlations will limit information, and it is far from clear that correlations would limit information enough to explain the behavior here.

Authors: We agree that this was a poor piece of writing – we have removed “individual”.

The second part of this comment is more substantial and is related to the concerns of Reviewer #2 about differential correlations. Please see the discussion before Responses to Reviewer #1.

7) In thinking about the neural basis of this behavior, the authors focus entirely on area MT. While I understand that there are many good reasons to consider MT as the primary candidate, especially for typical pursuit target sizes, one has to also consider the possibility that other areas may start to contribute more as target size becomes large. For example, with a 20 deg pursuit target, it may certainly be possible that there is now substantial involvement of the different subdivisions of area MST. If different areas are engaged to different degrees as target size changes, this would also factor into how one expects noise to grow with signal. The authors need to at least acknowledge this possibility and state the caveat that their modelling assumes that MT is the only relevant area, which may not be true.

Authors: The Discussion now considers the possible contributions of areas outside of MT and the likelihood that those areas receive much (but maybe not all) of their visual motion inputs from MT.

8) Lines 134-138: The variable $s_{\hat{}}$ here is described as the “sensory estimate”, but it is generally unclear in the paper whether the authors assume this to be the sensory estimate of target velocity or retinal velocity. This should be made clear in multiple places where velocity variables are used.

Authors: As discussed above, we consider $s_{\hat{}}$ to be an estimator during the open loop period of pursuit where retinal and target velocity are equivalent. We agree with the reviewer that a model of pursuit that generalizes to closed-loop would require explicit estimation of retinal motion. We have rewritten the text to clarify

9) Lines 157-160: Again, statistics are needed here.

Authors: As discussed in response to reviewer 1’s comments, we have included a bootstrap analysis to determine the significance of the fit improvement. The statistics validated our previous conclusions.

10) In the “biomimetic” model of MT, the authors model a neural population that has direction and speed tuning curves of fixed widths, and units with identical dynamic ranges. I found this to be a strange choice, since we know that diversity in a population of tuning curves has huge effects on how correlated noise limits information and what types of correlated noise limit information (e.g., Ecker et al 2011; Moreno-Bote et al. 2014). The authors’ formulation of the model is precisely the typical oversimplification for which standard limited-range correlations will limit information, whereas this will not be the case with diverse tuning curves. With diverse tuning curves, it will require differential correlations to limit information.

Authors: In response to this and reviewer 2's related concern we redid our simulations with variation in the shape and amplitude of the tuning curves of the model MT neurons. The key results do not change. These are the stimulations that now are reported in the paper.

As a side issue, Ecker et. al. (2011) makes the simplifying assumption that limited range correlations exist only along the axis of the stimulus parameter to be decoded (e.g. speed), whereas our model (and populations of real MT neurons) have noise correlations along several other key dimensions (e.g. direction, receptive field position). As discussed above, the added complexity of the covariance matrix makes it more likely that information will be limited (Moreno-Bote et. al. 2014), and accordingly that variance of the decoded output will be limited as well. As we state in the paper, including realistic and complex noise correlations in our model provides a parsimonious account of our results and those from perceptual psychophysics that observe no change in Weber fractions with increasing stimulus size (de Bruyn and Orban 1988; Verghese and Stone, 1995).

11) Directly following from the above point, it should be noted that the recorded populations were too small to do decoding without having to estimate the covariance matrix. And we know from the work of Pouget's group that estimating the covariance matrix is fraught with difficulties, and likely will misestimate the differential correlations. Given that we also don't know how the differential correlations in MT depend on stimulus size and how they are influenced by extra-retinal signals related to pursuit eye movements (which are known to modulate MT responses), the modelling results seem quite heavily dependent on a set of assumptions that are questionable. Consequently, I am not comfortable accepting the conclusions about sensory origins of noise when the model makes so many assumptions that may be inaccurate.

Authors: We think it is a dangerous proposition that the models that can be proposed should be limited by missing facts that cannot be measured practically such as correlations within large populations of MT neurons. Instead, we believe that a model should be judged by how well it fits with known results and its testable predictions. Our "biomimetic" model captures what is known about MT and, under these circumstances, it requires gain noise so that the model can reproduce 3 observations: (1) the quantitative magnitude of behavioral variance in speed estimates; (2) the magnitude of MT-pursuit correlations; and (3) the effects of changing target size on behavioral mean and variance. Further, our model makes specific and testable predictions about (1) the (lack of) dependence of correlations in MT on stimulus size and (2) the structure of the decoder downstream of MT. We have rewritten the text to be clear about these issues.

12) For the above reasons, it really seems important going forward to make population recordings in MT during this behavior to be able to constrain estimates of how the information in the MT population changes with target size, without the need for approximation of correlated noise. I am not saying that the authors need to do this for the manuscript to be reconsidered. But unless I am wrong in my concerns about a lot of the assumptions of the

model, I think the authors may have to really scale back and qualify their conclusions considerably.

Authors: We absolutely agree about the importance of this experiment. Right now, it is probably impossible to perform because of the limits of recording technology and it is an unfunded mandate. And it would require 2-3 years of recording and data analysis. We have tried to be appropriately circumspect in identifying the assumptions of our model and being clear about its limits.

13) Lines 187-189: Again, this behavior of the neural model seems to be based on the assumption that MT neurons represent target velocity, not retinal velocity.

Authors: Please see above for our discussion about open-loop pursuit. In our experiments and model, retinal velocity is target velocity.

14) Figure 5F: This result is pitched as providing a good match to the actual data of Fig. 5D. While I certainly agree that it is closer than the model without gain noise, the predicted data of Fig. 5F have a distribution that is clearly quite different from the actual data. So I think the language around this claim needs to be softened up.

Authors: Our revised model now captures the distribution of the data quite well when gain noise is included.

15) Lines 281-282: "Our findings suggest that the flexibility afforded by independent gain control pathways comes at a cost of increased variability associated with noise that arises through imperfect circuit computations." Isn't this trivially true? Any time you add a new signal into a computation, noise associated with that new signal is going to combine with noise in the other signals. So unless I am missing something here, this seems like a statement that could have easily been made a priori.

Authors: We have revised to clarify that our results support the idea that optimization models using gain control should consider the contribution of a noisy gain to the optimal policy.

16) Lines 308-311: Here, the authors indicate that they used a much larger fixation window for the 20 deg pursuit targets than for the smaller targets, despite the fact that the pursuit gain for large targets is much closer to unity. This was puzzling and made me wonder about what the monkeys actually do in pursuing these large targets, and why such large fixation windows are needed. It would be useful for the authors to describe the patterns of positional errors that the monkeys make. Since they said something about removing trials with saccades, does this mean that monkeys maintain large positional errors during the pursuit of large targets? Are those positional errors systematic?

Authors: Reviewer #1 had similar questions and they are answered in detail in our response to his review. The executive summary is: (1) we used large fixation windows because the large

targets are ambiguous about where the monkey needs to fixate and we didn't want to discourage monkeys when they thought they were "looking at the target"; (2) our results were the same when we considered only trials where the monkeys were fixating within 2 and 2.5 deg of the center of the target at the onset of target motion for monkey R and monkey X, respectively (see Figure R1); (3) monkeys initiated pursuit on essentially all trials, with fairly invariant latencies – occasionally they tracked without looking at the center of the target, but the eye movements were superficially the same when they did so.

17) Lines 326-328: I really did not follow the logic here. 667 MT neurons to represent the central 30 deg of the visual field? Is that supposed to be an estimate for what is actually in the brain? If so, it must be orders of magnitude too small. So the authors should clarify what they intend this statement to mean.

Authors: We have revised lines 326-328 to indicate that we chose the number of neurons in our simulations strategically: large enough numbers to be variance-limiting, but small enough to be computationally tractable. We note that we have changed the number of model neurons used in the main text to 1280.

Figure R1. Effect of fixation eccentricity at motion onset on variability and gain of pursuit responses to each target size. To assess the degree to which variation in fixation position at target onset influenced our results, we reanalyzed our data using trials with fixation within 2.5 and 2 deg of the fixation point for monkeys X and R, respectively. A) Variance in pursuit initiation by monkey X as a function of mean eye speed for right pursuit (top) and left pursuit (bottom). Green triangles, red circles, and black circles correspond to the data for the small, medium, and large targets, respectively. B) As in panel A, but for monkey R. C) Trial-by-trial eye speed as a function of target speed for monkey X. Dashed line plots unity. Solid lines plot the best fitting linear model for each target condition. Color and marker conventions as in panel A. D) As in panel C, but for monkey R. The analysis shows that our behavioral results do not critically depend on the degree to which the monkeys accurately fixated the target on motion onset.

Figure R2. Comparison of gain inferred by regression to the gain inferred from the relationship between the mean and variance of pursuit responses. Our analysis in Figure 2 provided an estimate of the gain for each size condition through regression. Equation (1) implies that we can also infer the gain from the relationship between the mean and variation in motor responses. We therefore fit the gain noise model using Equation (4), but allowed G to change as a function of the stimulus size. Panels A and B show the fit of the model to the mean and variance of pursuit responses by monkey X and R, respectively, to rightward trials (top) and leftward trials (bottom). Conventions as in Figure 1. We then compare the gains inferred from the noise model to that inferred from regression (i.e. Figure 2B and C) in panel C. There is substantial agreement between the two methods for inferring the gain. It should be noted, however, that an exact agreement should not be expected. Our gain noise model is based on the estimated speed, \hat{s} , which can be nonlinear with respect to speed, s . The gain inferred from regression, however, assumes linearity in estimation. Therefore, an exact agreement would only be expected if the monkeys linearly estimated speed (see Appendix A.2). Figure 2B and C suggest the assumption of linearity is not correct.

REVIEWER COMMENTS

Reviewer #1 (Remarks to the Author):

The authors have provided a detailed and thorough response that addresses all of my major concerns.

I'm not sure that I properly conveyed my (minor) concern about comparing models with different complexity, but this does not seem like it will affect any outcomes.

Reviewer #2 (Remarks to the Author):

I am satisfied with the authors' detailed response to my comments. I especially appreciate the inclusion of the variance derivation in the appendix. Rather than respond individually to each comment, I will just state that the revisions have improved the paper substantially and I have no critical comments except this one: The sentence in the discussion "However, noise correlations in MT also depend on direction preferences and the distance between receptive fields" and the sentence after that one are rather vague. To the extent that speed tuning, direction tuning, and location tuning are roughly orthogonal features, it is unclear why noise that depends on one feature will influence the decodability of other features. Therefore this argument should either be streamlined a bit or removed.

Reviewer #3 (Remarks to the Author):

Overall, the authors have done a responsible job of responding to the critiques, and they have cleared up some of my misconceptions. As a result, the manuscript is substantially improved. While that is all good, there are still a few problems that need to be addressed for the manuscript to be acceptable.

1) I still feel like the overall pitch of the main findings lacks the proper balance. One major point is that gain noise can account for the behavioral findings. I find that convincing and interesting. The other major point made is that the effects of target size on behavior cannot be explained by the sensory representation. As described in my previous review, this is where I have more problems. The evidence presented that there is no sensory contribution is rather weak. It relies on a biomimetic model to make this argument, but the model is quite simple and thus not very convincing. We don't really know how the speed tuning of MT neurons might be altered for neurons with receptive fields that are on the edge of the stimulus, nor what effect asymmetric surrounds might have on the speed tuning of such neurons. And we don't know how the sensory decoding might change with target size and be weighted across

neurons that are partially stimulated. The authors only use one simple decoder. Thus, it is very difficult to rule out that some decoding of a real population of MT neurons might produce similar effects of target size on variance. The authors say that the requisite experiments are impossible. I don't agree, but I am not suggesting that they need to do them either. What I'm asking is that the authors don't present their two claims with equal weighting. From the outset of the paper, they should soften their claims that there is no sensory contribution. I think that takes little away from the paper, and would be appropriately cautious.

2) Some of the arguments made to support the lack of a sensory substrate based on existing literature have real problems. For example, consider the following statement (lines 131-133): "Available physiological evidence rules out the possibility that the change in slope arises from a change in the speed estimate, s^{\wedge} . The response of individual MT neurons to targets of different size has little to no effect on the speed tuning preferences of MT neurons^{19, 49}." Reference 19 is a lesion/behavior study and I don't see how it has any data that shows this. Reference 49 looks at how speed tuning to the center depends on speed of the surround, but I don't see how it addresses the effect of target size which is relevant here. This raises some concerns about the authors' evaluation of the literature. I hope that maybe they just cited the wrong papers here, but all of these arguments and references are going to need to be checked carefully. I don't see where they cite any studies that directly examine how speed tuning in MT depends on size. And even if such studies were done, they almost certainly would have used stimuli centered on the RF, thus leaving open the issues mentioned above about how speed tuning might depend on asymmetric activation of RFs and surrounds.

3) In response to previous comments about need for statistics, the authors added some bootstrap analysis. But if I understand it correctly, it has a serious problem. The authors formed distributions of RMSE values from 1000 bootstraps and then applied a paired t-test to determine whether the means of the distributions were different. One can see that this approach is flawed because the number of bootstraps is arbitrary. If you had used one billion bootstraps and then performed the paired t-test on two distributions with a billion observations each, then clearly even tiny differences would be significant since the paired t-test will be quite sensitive to the number of data points. I think a more appropriate approach would be to determine confidence intervals from each of the bootstrap distributions and then assess whether these intervals overlap.

We thank all the reviewers for their previous comments which lead to a substantially improved manuscript. Please see below for our detailed responses (in blue) to the remaining comments.

Reviewer #1 (Remarks to the Author):

The authors have provided a detailed and thorough response that addresses all of my major concerns. I'm not sure that I properly conveyed my (minor) concern about comparing models with different complexity, but this does not seem like it will affect any outcomes.

We have used left-out data to perform our model comparisons, a standard method for comparing models of different complexity (Bishop, 2006).

Reviewer #2 (Remarks to the Author):

I am satisfied with the authors' detailed response to my comments. I especially appreciate the inclusion of the variance derivation in the appendix. Rather than respond individually to each comment, I will just state that the revisions have improved the paper substantially and I have no critical comments except this one: The sentence in the discussion "However, noise correlations in MT also depend on direction preferences and the distance between receptive fields" and the sentence after that one are rather vague. To the extent that speed tuning, direction tuning, and location tuning are roughly orthogonal features, it is unclear why noise that depends on one feature will influence the decodability of other features. Therefore this argument should either be streamlined a bit or removed.

The reviewer points out ambiguity in our discussion of the structure of noise correlations and its relationship to decodability. While perhaps counter-intuitive, it is trivial to create a situation where correlations structured with respect to an orthogonal feature will limit decoding accuracy of the feature of interest. Consider two neurons, both with linearly increasing firing rates with respect to speed and positive correlations because they have similar direction preferences. When both neurons increase their firing, was the increase due to an increase in speed or correlated noise? In fact, these neurons have correlation structure that is information limiting as defined by Moreno-Bote et. al. (2014). It is not material if the correlations are structured w.r.t. speed or direction in this case, only that they exist and are positive.

In the more realistic case with a large population of neurons and varied tuning curves, the important point to consider is whether additional structure in the noise correlations results in additional significant modes in the noise correlation matrix; even one additional mode will allow noise structures that are not differential to be information limiting (Moreno-Bote et. al. (2014)). If there is structure along orthogonal features, building this into the correlation matrix results in additional significant modes. To understand why, consider building a correlation matrix based on only the difference in speed preferences. If the matrix is ordered by speed preference, the correlation structure will smoothly decrease with the distance from the diagonal of the matrix. Now consider adding correlations based on the difference in direction preferences. If you secondarily order the matrix according to direction preference, this correlation structure will appear as blocks off the diagonal. Intuitively, these blocks will increase the rank of the correlation matrix, with additional high-value eigenvalues associated with the new modes.

Unfortunately, this fact was lost in our attempt to succinctly describe the effect. We have revised the two sentences in the discussion to clarify as well as streamline the argument.

Reviewer #3 (Remarks to the Author):

Overall, the authors have done a responsible job of responding to the critiques, and they have cleared up some of my misconceptions. As a result, the manuscript is substantially improved. While that is all good, there are still a few problems that need to be addressed for the manuscript to be acceptable.

We thank the reviewer for reconsidering our manuscript in light of our clarifications.

1) I still feel like the overall pitch of the main findings lacks the proper balance. One major point is that gain noise can account for the behavioral findings. I find that convincing and interesting. The other major point made is that the effects of target size on behavior cannot be explained by the sensory representation. As described in my previous review, this is where I have more problems. The evidence presented that there is no sensory contribution is rather weak. It relies on a biomimetic model to make this argument, but the model is quite simple and thus not very convincing. We don't really know how the speed tuning of MT neurons might be altered for neurons with receptive fields that are on the edge of the stimulus, nor what effect asymmetric surrounds might have on the speed tuning of such neurons. And we don't know how the sensory decoding might change with target size and be weighted across neurons that are partially stimulated. The authors only use one simple decoder. Thus, it is very difficult to rule out that some decoding of a real population of MT neurons might produce similar effects of target size on variance. The authors say that the requisite experiments are impossible. I don't agree, but I am not suggesting that they need to do them either. What I'm asking is that the authors don't present their two claims with equal weighting. From the outset of the paper, they should soften their claims that there is no sensory contribution. I think that takes little away from the paper, and would be appropriately cautious.

We would like to point out that results from perceptual psychophysics also support our conclusion that the contribution of the sensory system to the changes in Weber fractions is limited. Still, the reviewer's point that there are a substantial number of known-unknowns that, once known, might force us to revise our conclusions is valid. While we have tried to be appropriately circumspect regarding our model assumptions, our language throughout the manuscript does a disservice to these unknowns. We have tempered our language throughout the manuscript and have added a paragraph to the results to that (1) summarizes our attempts to mitigate the impact of unknown response properties and (2) directly states how future results measuring the physiology of MT neurons in our task may invalidate our findings.

2) Some of the arguments made to support the lack of a sensory substrate based on existing literature have real problems. For example, consider the following statement (lines 131-133): "Available physiological evidence rules out the possibility that the change in slope arises from a change in the speed estimate, s^{\wedge} . The response of individual MT neurons to targets of different size has little to no effect on the speed tuning preferences of MT neurons^{19, 49}." Reference 19 is a lesion/behavior study and I don't see how it has any data that shows this. Reference 49 looks at how speed tuning to the center depends on speed of the surround, but I don't see how it addresses the effect of target size which is relevant here. This raises some concerns about the authors' evaluation of the literature. I hope that maybe they just cited the wrong papers here, but all of these arguments and references are going to need to be checked carefully. I don't see where they cite any studies that directly examine how speed tuning in MT depends on size. And even if such studies were done, they almost certainly would have used stimuli centered on the RF, thus leaving open the issues mentioned above about how speed tuning might depend on asymmetric activation of RFs and surrounds.

The reviewer is correct that we have made an error in our citation to Newsome et. al. (1985) here. Our citation of Born (2002) was intentional, but thanks to the reviewer we now recognize this sentence poorly reflects the experiment and findings in Born (2002). Figure 10E of Born (2002) performs a quantitative analysis of the

preferred speed of MT neurons when there was motion in the surround and “shifts of the peak of the speed tuning curve induced by surround motion were clustered around zero,” suggesting speed tuning is not changed by surround stimulation. We have revised the sentence to more accurately reflect that reference’s findings and our interpretation of how they will generalize to targets of different size.

In addition, we have now carefully reviewed all our citations to ensure that they all correctly refer to the literature.

3) In response to previous comments about need for statistics, the authors added some bootstrap analysis. But if I understand it correctly, it has a serious problem. The authors formed distributions of RMSE values from 1000 bootstraps and then applied a paired t-test to determine whether the means of the distributions were different. One can see that this approach is flawed because the number of bootstraps is arbitrary. If you had used one billion bootstraps and then performed the paired t-test on two distributions with a billion observations each, then clearly even tiny differences would be significant since the paired t-test will be quite sensitive to the number of data points. I think a more appropriate approach would be to determine confidence intervals from each of the bootstrap distributions and then assess whether these intervals overlap.

The reviewer’s observation is quite astute and points to a lack of clarity in our report of the methods. Indeed, a standard t-test as implemented in a statistical package (e.g. MATLAB’s `ttest`) would be subject to this error in a bootstrap analysis. However, we are well aware that the t-test, as traditionally formulated, is dependent on the number of observations, N . To avoid allowing the bootstrap procedure to find significance at an arbitrary N , we calculated our t-statistic as the mean difference (across bootstraps) divided by the *standard deviation* (across bootstraps). Importantly, the standard deviation of the bootstrap distribution serves as an estimate of the standard error of the true distribution of differences. Because the standard deviation is stable with increasing N , our implementation of the t-test does not suffer from the error described by the reviewer. We have revised our methods to clarify how the t-statistic was calculated so that our methods are now clear. Finally, we note that, because our t-statistic is the mean over the standard deviation of the bootstrap distribution, the confidence intervals of the distribution can be inferred by the reader.

REVIEWERS' COMMENTS

Reviewer #3 (Remarks to the Author):

The authors have made good efforts to address my remaining concerns, and have now couched their findings appropriately relative to the uncertainty of some of their assumptions. I now find the overall tone and pitch of the paper more appropriate. They have also clarified that my concern about the bootstrap analysis was not valid given how the test was performed.

I have no additional concerns.

Reviewer #3 (Remarks to the Author):

The authors have made good efforts to address my remaining concerns, and have now couched their findings appropriately relative to the uncertainty of some of their assumptions. I now find the overall tone and pitch of the paper more appropriate. They have also clarified that my concern about the bootstrap analysis was not valid given how the test was performed.

I have no additional concerns.

We thank Reviewer #3 for their effort and believe their comments have been helpful for us to improve the tone of the manuscript and reproducibility of the methods we used to evaluate the significance of our findings. We would again like to thank all three reviewers for their effort and insightful comments.